# Large Language Models Meet Symbolic Provers for Logical Reasoning Evaluation

**Chengwen Qi**[1,2*]  **Ren Ma**[2*]  **Bowen Li**[2*]  **He Du**[2,3]  **Binyuan Hui**  **Jinwang Wu**[1]
**Yuanjun Laili**[1,4,5†]  **Conghui He**[2†]
[1]Beihang University  [2]Shanghai Artificial Intelligence Laboratory
[3]Fudan University  [4]Zhongguancun Laboratory
[5]State Key Laboratory of Intelligent Manufacturing Systems Technology, Beijing
{Chengwen_qi, lailiyuanjun}@buaa.edu.cn
{maren, libowen, heconghui}@pjlab.org

## Abstract

First-order logic (FOL) reasoning, which involves sequential deduction, is pivotal for intelligent systems and serves as a valuable task for evaluating reasoning capabilities, particularly in chain-of-thought (CoT) contexts. Existing benchmarks often rely on extensive human annotation or handcrafted templates, making it difficult to achieve the necessary complexity, scalability, and diversity for robust evaluation. To address these limitations, we propose a novel framework called ProverGen that synergizes the generative strengths of Large Language Models (LLMs) with the rigor and precision of symbolic provers, enabling the creation of a scalable, diverse, and high-quality FOL reasoning dataset, ProverQA. ProverQA is also distinguished by its inclusion of accessible and logically coherent intermediate reasoning steps for each problem. Our evaluation shows that state-of-the-art LLMs struggle to solve ProverQA problems, even with CoT prompting, highlighting the dataset's challenging nature. We also finetune Llama3.1-8B-Instruct on a separate training set generated by our framework. The finetuned model demonstrates consistent improvements on both in-distribution and out-of-distribution test sets, suggesting the value of our proposed data generation framework. Code available at: `https://github.com/opendatalab/ProverGen`

## 1 Introduction

First-order logic (FOL) reasoning, which involves sequential deduction and the application of facts and logical rules to derive conclusions or make decisions, is a fundamental aspect of human intelligence (Saparov & He, 2022; Pan et al., 2023). The complexity of performing multi-step reasoning while maintaining logical coherence across these steps makes FOL reasoning an ideal testbed for evaluating the reasoning capabilities of Large Language Models (LLMs), particularly in scenarios such as Chain-of-Thought (CoT) prompting (Wei et al., 2022) and planning (Wang et al., 2024; Valmeekam et al., 2022). An effective benchmark for FOL reasoning should satisfy several key criteria: (1) *Scalability*, enabling expansion in both volume and complexity with minimal manual intervention; (2) *Natural and Diverse Language*, capturing a wide range of natural language expressions to reflect real-world linguistic variability; (3) *Symbolic Representations*, providing formal symbolic structures that can be validated through automated symbolic provers, ensuring dataset integrity and supporting downstream tasks like NL-FOL translation and tool based logic problem solving (Pan et al., 2023; Olausson et al., 2023); and (4) *Faithful Reasoning Chains*, where each instance includes intermediate reasoning steps clearly articulated in both symbolic and natural language formats, promoting transparency and facilitating further model training.

However, existing FOL reasoning datasets only partially fulfill these requirements (see Table 1 for more details). Template-based datasets such as ProntoQA (Saparov & He, 2022; Saparov et al., 2024), ProofWriter (Tafjord et al., 2021), and RuleTaker (Clark et al., 2021) are scalable but often

---

*Equal contribution
†Corresponding authors

lack diversity and rely on overly simplistic rules. On the contrary, manually curated datasets like FOLIO (Han et al., 2022) offer rich diversity in natural language expressions and more complex logic rules, but they are limited in size due to the extensive human effort needed for creation. Additionally, data contamination is also a significant issue in manually annotated datasets, as it is difficult to update them frequently. This limitation can lead to biased evaluations and hinder true generalization due to potential data leakage. Another common limitation across previous datasets is the lack of well-defined and easily accessible reasoning chains. FOLIO, for instance, does not provide any reasoning chains, while ProofWriter and RuleTaker include reasoning chains, but they are not readily accessible and require users to implement additional code to extract them for each problem.

Table 1: Comparison of existing FOL reasoning datasets with ProverQA.

| Dataset | Creation Method | Scalability | Natural & Diverse Language | Symbolic Representations | Faithful Reasoning Chains |
|---|---|---|---|---|---|
| RuleTaker | Synthetic | ✓ | ✗ | ✗ | ✓ |
| ProofWriter | Synthetic | ✓ | ✗ | ✗ | ✓ |
| ProntoQA | Synthetic | ✓ | ✗ | ✓ | ✓ |
| ProntoQA-OOD | Synthetic | ✓ | ✗ | ✓ | ✓ |
| LogicNLI | Synthetic | ✓ | ✗ | ✗ | ✗ |
| FOLIO | Manual | ✗ | ✓ | ✓ | ✗ |
| ProverQA | Synthetic | ✓ | ✓ | ✓ | ✓ |

To address these gaps, we propose a novel framework that leverages the strengths of LLMs and symbolic provers to generate an extensible, diverse, and high-quality dataset for FOL reasoning. While LLMs have been increasingly utilized for synthesizing datasets across various tasks (Taori et al., 2023; Acharya et al., 2023; Gajjar & Shah, 2024), they fall short in producing reliable reasoning, often lacking faithfulness and transparency (Golovneva et al., 2022; Ribeiro et al., 2022; Lyu et al., 2023), particularly when it comes to long reasoning chains. On the other hand, symbolic provers are known for their reliability and precision in executing complex logical inferences. By combining the generative capabilities of LLMs with the rigor of symbolic provers, our framework offers a robust solution for producing FOL reasoning data that ensures both diversity and logical soundness. It also mitigates the data contamination challenge by enabling the generation of new datasets using diverse models and controlled complexity, ensuring that the datasets remain fresh and uncontaminated. Our framework follows a structured three-step pipeline: (1) we first use an LLM to generate a unique background story for each problem based on a given subject and a characteristic keyword; (2) we employ a symbolic prover (i.e., Prover9) to construct reasoning trees and compute the truth values of relevant facts; and (3) we utilize an LLM to translate these logical expressions into natural language statements.

We use Llama3.1-70B-Instruct within our proposed generation framework to create a new FOL reasoning benchmark, **ProverQA**, consisting of 1,500 instances across three difficulty levels: *easy*, *medium*, and *hard*. We evaluate ProverQA on several state-of-the-art (SOTA) LLMs and find that even with CoT prompting, models barely exceed 50% accuracy on the hard subset, highlighting the significant challenge posed by ProverQA. ProverQA's difficulty settings are carefully designed to ensure appropriate complexity, avoiding both performance saturation and uninformative underperformance (e.g., random guessing), enabling accurate evaluation of both powerful and smaller language models' reasoning abilities. To further explore the potential of our proposed generation framework, we finetune Llama3.1-8B-Instruct on a separately generated ProverQA training set and observe substantial performance gains on in-distribution (ID) evaluations. Remarkably, this finetuned model also shows consistent improvements on out-of-distribution (OOD) test sets, significantly outperforming the same model finetuned on other existing FOL training sets.

## 2 RELATED WORKS

**Logical Reasoning Datasets** Logical reasoning is a fundamental ability of intelligent systems and researchers have proposed a number of datasets to evaluate logical reasoning abilities of language models. These datasets can be divided into two groups: manually crafted ones and automatically

generated ones. Human-crafted datasets are typically sourced from existing examinations (Yu et al., 2019; Liu et al., 2021; 2023; Zhong et al., 2021) or directly written by domain experts (Hu et al., 2023; Han et al., 2022). Although they often exhibit higher quality and complexity, they also suffer from limited quantity and high creation costs. Automatically generated datasets are often created using rules and predefined natural language templates (Sinha et al., 2019; Hahn et al., 2021; Clark et al., 2021; Tafjord et al., 2021; Tian et al., 2021; Saparov & He, 2022; Saparov et al., 2024; Sun et al., 2021). While these datasets enable automatic problem generation, they often lack linguistic diversity and are constrained by rigid, monotonous rules. None of them cover all aspects of first-order logic, and they cannot be used to generate problems with more complex rules.

**Symbolic Prover Augmented LLMs** Symbolic provers are computer programs used to generate proofs and solutions of mathematical theorems automatically. In logical reasoning problems, symbolic provers, such as SymPy (Meurer et al., 2017), Z3 (De Moura & Bjørner, 2008), Lean (De Moura et al., 2015), Pyke (Frederiksen, 2008) and Prover9 (McCune, 2005), stand out with faithful solutions and traceable intermediate processes. Recent works (Pan et al., 2023; Olausson et al., 2023; Ye et al., 2024; Zhang et al., 2024) combines LLMs and symbolic provers to solve logic reasoning problems. They use LLMs to translate problems from natural language into symbolic representations and then solve that problem with a symbolic prover. In contrast to previous works that utilize symbolic provers solely as problem solvers, our approach integrates symbolic provers into the generation process of first-order logic (FOL) problems. This integration allows us to bypass the intricate interrelations and couplings among logical relations, enabling the generation of diverse rules and their correct chaining to form comprehensive problems.

## 3 METHODOLOGY

### 3.1 TASK FORMULATION

Given a set of premises $\mathcal{P}$, consisting of facts $\mathcal{F}$ and rules $\mathcal{R}$, denoted as $\mathcal{P} = (\mathcal{F}, \mathcal{R})$, the goal of FOL reasoning is to determine whether a given goal $\mathcal{G}$ (which can be a fact or a rule) is True, False, or Uncertain. Specifically, $\mathcal{F} = \{f_1, f_2, \cdots, f_m\}$ represents a set of facts, where each fact $f_i$ ($1 \leq i \leq m$) is a declarative statement describing attributes or characteristics of subjects. For example, `Sawyer has good dance skills` is a fact that specifies the attribute (`has good dance skills`) of the subject (`Sawyer`). Similarly, $\mathcal{R} = \{r_1, r_2, \cdots, r_n\}$ comprises a set of rules, where each rule $r_j$ ($1 \leq j \leq n$) defines relationships between multiple facts using seven FOL symbols: conjunction ($\wedge$), disjunction ($\vee$), negation ($\neg$), implication ($\rightarrow$), equivalence ($\equiv$) [1], universal quantifier ($\forall$), and existential quantifier ($\exists$). For example, the rule `If Sawyer has good dance skills and has charisma, then he is a successful dancer` illustrates a relationship between two facts and their conclusion. It is worth noting that some existing datasets, such as ProntoQA (Saparov & He, 2022) and ProofWriter (Tafjord et al., 2021), do not fully encompass all seven FOL relationships, whereas our proposed framework ensures complete coverage.

### 3.2 LLM & PROVER-EMPOWERED GENERATION FRAMEWORK

Our proposed generation framework, which integrates the generative capabilities of LLMs with the precision of a symbolic prover, generates FOL reasoning problems through a structured three-stage process, as illustrated in Figure 1. First, a background story is created for each problem using sampled subject names and keywords to ensure linguistic diversity. In the next stage, the logic skeleton generation employs a symbolic prover (i.e., Prover9) to construct reasoning trees and determine the truth values of the relevant facts. Finally, the statement translation phase uses LLMs to convert these logical expressions, including both premises and goals, into natural language, ensuring clarity and coherence. An example of a generated FOL problem is shown in Figure 2.

#### 3.2.1 BACKGROUND STORY GENERATION

Achieving naturalness in language expression is crucial for creating a plausible FOL dataset. Existing approaches often rely on extensive human annotation or rigidly structured templates. For in-

---

[1] In line with previous datasets, we replace equivalence ($\equiv$) with exclusive disjunction ($\oplus$), which can be interpreted as a combination of equivalence ($\equiv$) and negation ($\neg$): $f_1 \oplus f_2 = \neg(f_1 \equiv f_2)$

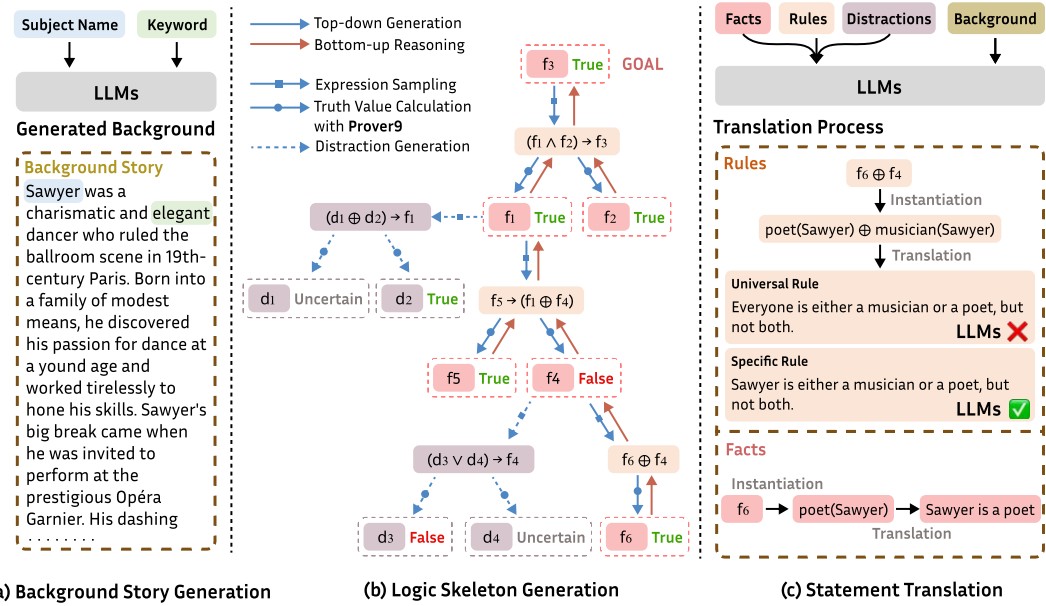

Figure 1: An overview of our ProverGen framework. (a) **Background Story Generation** (Section 3.2.1). Given a subject, `Sawyer`, and a seed keyword, `elegant`, LLMs generate a background story to establish context and ensure linguistic diversity. (b) **Logic Skeleton Generation** (Section 3.2.2). A top-down approach is used to generate the logic skeleton, forming the reasoning tree for the FOL problem. This involves two iterative steps: expression sampling and truth value calculation using the Prover9 prover. Distractions are also incorporated to test the robustness of model's reasoning capabilities. (c) **Statement Translation** (Section 3.2.3). LLMs translate the facts and rules from the logic skeleton into natural language, guided by the previously generated background story. Each rule is translated into a universal version and a specific version. The universal version is preferred if it does not contradict with common sense, otherwise the specific one is selected.

stance, PrOntoQA (Saparov & He, 2022) and ProofWriter (Tafjord et al., 2021) employ handcrafted templates, which result in datasets lacking linguistic diversity and naturalness.[2]

In contrast, FOLIO (Han et al., 2022) retrieves documents from Wikipedia and manually extracts facts and rules. However, this process requires substantial human effort and is susceptible to syntactic typos and reasoning errors introduced during annotation, as observed in FOLIO. Additionally, models may exploit *shortcuts* by leveraging pre-trained knowledge to predict answers directly, bypassing rigorous logical reasoning (Qi et al., 2023; Geirhos et al., 2020), since the facts and rules in the dataset are often derived from the real-life.

To efficiently synthesize plausible natural FOL problems, we leverage LLMs to generate a unique background story for the subject of each problem, which serves to instantiate the statements (both premises and goals) within the problem. However, simply prompting LLMs to generate stories without any guidance can be suboptimal, as they struggle to produce diversified content with high quality, as noted by Jentzsch & Kersting (2023); Sudalairaj et al. (2024). To mitigate these issues, we prepare a characteristic keyword for the subject within the problem to guide the generation. Specifically, the process begins by assigning a unique name to the subject, which is randomly sampled

---

[2]Both ProofWriter and PrOntoQA generate natural language expressions by using predefined templates for each logical rule. For example, for a rule like $A \rightarrow B$, they may define a template such as `Every A is B`, and then randomly select words to replace $A$ and $B$. PrOntoQA intentionally avoids real-world concepts to prevent conflicts with real-world knowledge. They also evaluate LLMs' behavior on examples that contain facts and rules that are consistent (or inconsistent) with the real-world knowledge. In contrast, ProofWriter does not incorporate such a mechanism.

from publicly accessible datasets of human names[3] and pet names[4]. Next, we sample a keyword from WordNet (Miller, 1995) to serve as a defining characteristic or attribute of the subject, ensuring semantic diversity in the generated content. These elements, the name and the characteristic keyword, are then integrated into the prompt to guide LLMs to generate a contextually rich and varied background story. This controlled generation helps promote diversity in the dataset. The detailed prompt used in our experiments is outlined in Appendix A.

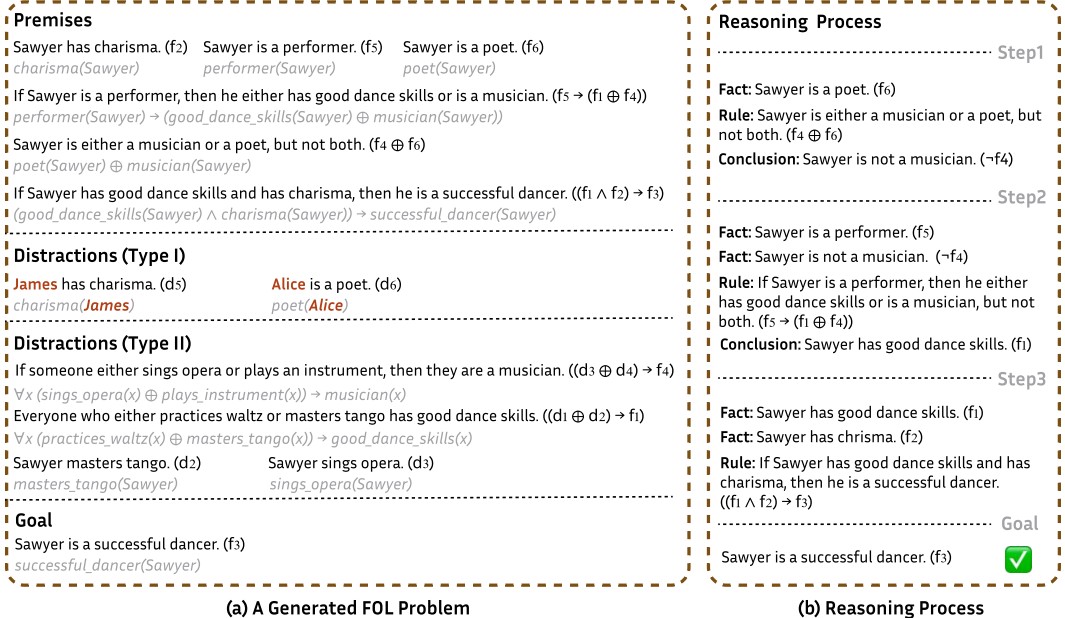

Figure 2: A generated FOL problem (a) with its corresponding reasoning process (b).

### 3.2.2 Logic Skeleton Generation

Logical soundness is a fundamental requirement in constructing a robust FOL reasoning dataset. While LLMs are highly proficient at generating natural language, they often fall short in maintaining logical coherence, particularly when dealing with intricate, multi-step reasoning processes. To address this limitation, we introduce a novel approach that integrates symbolic prover into the data generation pipeline, allowing for the construction of logically rigorous FOL reasoning problems.

In our data generation framework, we start with constructing a logic skeleton for each problem, represented as a reasoning tree with the ultimate goal at its root (as shown in Figure 1(b)). Previous works typically generate reasoning problems in a *bottom-up* approach by starting with premises and building toward conclusions. We argue that this approach could be flawed as it can often result in disconnected or contradictory premises, making it difficult to ensure that all facts are logically coherent with each other. For instance, if the problem includes facts such as $f_1$: `Sawyer is a poet`, $f_2$: `Sawyer comes from Mars`, and $f_3$: `Sawyer is a large, fluffy Maine Coon cat`, then integrating these elements into a consistent reasoning chain is challenging. Therefore, our framework adopts a novel *top-down* approach, which begins with the goal and works backward to derive the relevant premises and rules needed to logically support that goal. This method ensures that each fact and rule contributes directly to the reasoning process, improving both the coherence and logical soundness of the generated problems. Moreover, this top-down structure simplifies the problem of logical consistency by allowing LLMs to focus on facts that are directly relevant to the conclusion, without needing to reconcile them with unrelated information.

More specifically, as shown in Figure 1(b), we begin by setting the truth value of the goal, denoted as $f_3 = \text{True}$, and treat this as the root node of the reasoning tree. Next, we sample a logical expression

---

[3] https://github.com/aruljohn/popular-baby-names
[4] https://github.com/fregante/pet-names/blob/master/pet-names.json

for this node, $(f_1 \wedge f_2) \rightarrow f_3$, which establishes a relationship between the goal $f_3$ and the premises $f_1$ and $f_2$. At this stage, we use the symbolic prover Prover9 to calculate the truth values of $f_1$ and $f_2$, ensuring that the premises logically support the goal. Once $f_1$ and $f_2$ are assigned truth values, they become the current nodes in the reasoning tree. For each of these nodes, we continue the process by sampling new logical expressions. For instance, for $f_1 = \mathsf{True}$, we sample a new rule $f_5 \rightarrow (f_1 \oplus f_4)$, where $f_5$ and $f_4$ are additional facts to be evaluated. We then calculate the truth values for these new premises using Prover9, and they become the next current nodes in the tree. This iterative process is repeated for each current node, expanding the reasoning tree step by step. The process continues until we have generated the desired number of reasoning steps, ensuring that each layer of the reasoning tree is logically consistent and fully developed, forming a coherent and complete logical structure for the problem.

To further assess the robustness of models in handling FOL reasoning, we introduce distractions into the generated problems. Distractions, denoted as $d$, are an additional set of facts and rules not essential to solving the problem. The inclusion of distractions forces models to distinguish between relevant and distracting information. We implement two types of distractions in our framework. The first type follows existing works (Tafjord et al., 2021; Saparov & He, 2022), where we randomly introduce facts or rules that are related to the problem's domain but use a different subject name (see Figure 2). The second, more complex type of distraction leverages the power of symbolic provers to introduce facts and rules that are semantically or logically related to the reasoning chain, but which lead to an Uncertain judgment. This type of distraction requires models to reason over logically relevant information that does not directly impact the truth value of the core facts needed to reach the final conclusion. For instance, suppose $f_1$ is a critical fact in the reasoning tree, and we introduce two distractions, $d_1 = \mathsf{True}$ and $d_2 = \mathsf{Uncertain}$, with the rule $(d_1 \oplus d_2) \rightarrow f_1$. Here, the truth value of $f_1$ remains Uncertain because the rule involves incomplete information. Models must reason through these distractions, yet ultimately recognize that they do not alter the critical facts that drive the reasoning chain forward.

### 3.2.3 STATEMENT TRANSLATION

After generating the logical skeleton of a problem, we use LLMs to translate each rule and fact into natural language within the framework of the background story generated in Section 3.2.1. As illustrated in Figure 1(c), we first prompt LLMs to instantiate placeholders (such as $f_6$ and $f_4$) in the logic expressions with appropriate predicates, like `poet` and `musician`. Once the placeholders are instantiated, the logic expressions are translated into natural language. To maintain consistency throughout the translation process, we provide the previously translated facts and rules as references in the prompt for subsequent translations (details of the translation prompt can be found in Appendix B).

To ensure the plausibility and soundness of the rules, we instruct LLMs to generate two versions of each rule: a universal and a specific version. The universal rule refers to statements that apply to everyone or everything, such as `Everyone is either a musician or a poet, but not both`. The specific rule, on the other hand, applies to an individual subject, such as `Sawyer is either a musician or a poet, but not both`. LLMs then assess whether the universal rule aligns with common sense. If it does, we use the universal version; if not, we opt for the specific rule. For instance, the universal rule in Figure 1(c) does not align with common sense (as there are more professions than just musicians and poets), so we choose the specific version of the rule. The final stage of the proposed framework involves assembling the facts, rules, and distractions generated in previous stages into a complete FOL problem. An example of the constructed problem with its corresponding reasoning steps is provided in Figure 2.

### 3.3 THE PROVERQA BENCHMARK

We utilize Llama3.1-70B-Instruct throughout our proposed generation framework to develop a new FOL reasoning benchmark, **ProverQA**, comprising 1,500 instances. The benchmark is evenly divided into three subsets based on the length of the reasoning chains: *easy* (1-2 steps), *medium* (3-5 steps), and *hard* (6-9 steps). In addition to the number of reasoning steps, the *hard* subset is characterized by more diverse reasoning directions. Reasoning in the *easy* and *medium* subsets typically follows a straightforward left-to-right progression. For example, given $f_1 = \mathsf{True}$ and $f_1 \rightarrow f_2$, the model can deduce that $f_2 = \mathsf{True}$. In contrast, the *hard* subset involves more complex patterns,

requiring models to infer $f_1 = $ False given $f_2 = $ False and $f_1 \rightarrow f_2$, thus demonstrating a more intricate reasoning process. Each problem in the benchmark contains two types of distractions (see Section 3.2.2), where the number of distractions for each type is randomly sampled. Moreover, to further challenge the robustness of models, we shuffle the order of the premises within each problem, ensuring that models cannot exploit fixed patterns or sequences to derive conclusions. Detailed examples of the ProverQA are provided in Appendix D.

## 4 EVALUATION ON THE PROVERQA

### 4.1 EXPERIMENTAL SETUP

We evaluate current SOTA proprietary LLMs (`GPT-4o` (OpenAI, 2024) and `Claude3.5` (Anthropic, 2024))[5] and open-source LLMs (`Mistral-7B-Instruct`, `Mistral-Large-Instruct`, `Mixtral-8x22B-Instruct-v0.1` (Jiang et al., 2023; 2024) and `Llama3.1-8B-Chat`, `Llama3.1-70B-Chat` (Dubey et al., 2024)) on the ProverQA. We utilize two prompting strategies: *Standard* and *CoT*. *Standard* prompting uses 2-shot in-context learning to prompt LLMs to answer questions directly, whereas *CoT* prompting employs 2-shot examples to instruct the model to solve questions step by step. Examples of both prompting strategies can be found in Appendix E.

### 4.2 MAIN RESULTS

Table 2: The performance of LLMs on ProverQA and existing FOL reasoning datasets. †: The average performance improvement of *CoT* prompting over *Standard* prompting across models on each evaluation set, which indicates the average gain *CoT* prompting brings on each set.

| | ProverQA | | | ProntoQA | ProofWriter | FOLIO | Avg |
|---|---|---|---|---|---|---|---|
| | Easy | Medium | Hard | | | | |
| *Standard Prompting* | | | | | | | |
| GPT-4o | 87.20 | 68.60 | 46.20 | 91.80 | 56.33 | 67.86 | 69.66 |
| Claude-3.5-Sonnet | 85.00 | 68.20 | 42.80 | 88.60 | 55.00 | 77.85 | 69.58 |
| Llama3.1-8B-Instruct | 46.60 | 43.00 | 39.00 | 50.40 | 43.80 | 54.29 | 46.18 |
| Llama3.1-70B-Instruct | 82.00 | 64.20 | 47.60 | 80.60 | 50.33 | 67.86 | 65.28 |
| Mistral-7B-Instruct | 56.80 | 46.80 | 37.20 | 50.00 | 42.33 | 54.29 | 47.90 |
| Mistral-Large-Instruct | 84.60 | 69.20 | 49.60 | 71.00 | 60.33 | 77.14 | 68.65 |
| Mixtral-8x22B-Instruct | 75.40 | 57.40 | 39.00 | 65.20 | 40.17 | 74.29 | 58.58 |
| *CoT Prompting* | | | | | | | |
| GPT-4o | 94.20 | 79.40 | 50.00 | 100.00 | 67.33 | 72.14 | 77.18 |
| Claude-3.5-Sonnet | 95.20 | 83.60 | 56.40 | 99.20 | 76.33 | 80.71 | 81.91 |
| Llama3.1-8B-Instruct | 75.60 | 46.60 | 33.60 | 79.60 | 56.83 | 63.57 | 59.30 |
| Llama3.1-70B-Instruct | 90.40 | 73.20 | 46.80 | 95.40 | 71.17 | 74.29 | 75.21 |
| Mistral-7B-Instruct | 72.00 | 51.00 | 41.80 | 61.20 | 46.00 | 63.58 | 55.93 |
| Mistral-Large-Instruct | 92.60 | 75.80 | 52.20 | 98.60 | 73.50 | 83.57 | 79.38 |
| Mixtral-8x22B-Instruct | 87.60 | 66.80 | 47.60 | 79.60 | 57.67 | 73.57 | 68.81 |
| Avg $\Delta^\dagger$ | 12.86 | 8.43 | 3.86 | 16.57 | 14.36 | 5.41 | 10.25 |

**ProverQA as a New Challenge** Overall, model performance on ProverQA declines as problem difficulty increases, with the lowest scores observed on the hard subset *across all existing datasets* as seen in Table 2. Specifically, strong models such as GPT-4o, Claude-3.5-Sonnet and Mistral-Large-Instruct barely exceed 50% accuracy on the hard subset, even with CoT prompting. This result

---

[5]The version of `GPT-4o` we used in our experiments is `gpt-4o-2024-08-06`, and the version of `Claude3.5` we used is `claude-3-5-sonnet-20240620`.

indicates that even the most advanced LLMs perform poorly on sufficiently complex problems with long reasoning chains. In contrast, simpler benchmarks like ProntoQA appear to be nearly solved by models like GPT-4o and Claude-3.5-Sonnet. However, smaller models still struggle to achieve satisfactory performance, even on the easy subset of ProverQA, suggesting that these models are not yet equipped to handle FOL reasoning effectively.

**Comparison of prompting strategies**  Compared with *Standard* prompting, *CoT* prompting notably enhances the performance of models on easy datasets but does not yield a similar improvement on harder datasets, as presented in the last row of Table 2. The improvement on the easy dataset is more pronounced because performance on easy problems is probably limited by the CoT capability, whereas the bottleneck for hard problems may involve other factors beyond CoT capability. We have also conducted experiments using 5-shot examples; the results can be found in Appendix G.

### 4.3    ABLATION STUDY ON DISTRACTIONS AND PREMISE ORDER

By default, the ProverQA benchmark includes two types of distractions and shuffles the order of premises within each problem (see Section 3.3). To assess how these two factors influence the evaluation results of LLMs, we conduct ablation studies using GPT-4o with *CoT* prompting in this section. Specifically, in the first setting, we remove all distractions from the problems to observe how the absence of irrelevant information affects the model's reasoning performance. Results in Table 3 exhibit that the model performs better in this situation. In the second setting, we *additionally* reorder the premises to match the logical progression of the underlying reasoning chain. This turns out that the model performs even better when premises are presented in a coherent sequence. Consequently, these findings suggest both factors are valuable for constructing a robust FOL reasoning benchmark for LLMs.

Table 3: Performance of GPT-4o on different variants of ProverQA using CoT prompting. The first row represents the default ProverQA setting, which features distractions and shuffled premises.

| Distractions | Shuffled Premises | Easy | Medium | Hard |
|:---:|:---:|:---:|:---:|:---:|
| ✓ | ✓ | 94.20 | 79.40 | 50.00 |
| ✗ | ✓ | 94.40 | 82.20 | 57.40 |
| ✗ | ✗ | 95.60 | 87.40 | 62.40 |

## 5    FINETUNING WITH PROVERQA TRAIN DATA

Beyond its role as an evaluation benchmark, we also investigate whether the training data synthesized by our proposed framework can improve a model's FOL reasoning capabilities, particularly in out-of-distribution (OOD) scenarios. To this end, we generate a dedicated training set and finetune an open-source model, Llama3.1-8B-Instruct. For comparison, we also finetune the same model on existing FOL reasoning datasets, such as ProofWriter and FOLIO. We exclude ProntoQA from this experiment due to the absence of a publicly available training set. Detailed training configurations are provided in Appendix F.

### 5.1    EVALUATION RESULTS

**In-Distribution (ID) Performance**  As shown in Table 4, finetuning on the ProverQA training set results in a substantial performance improvement across all subsets of the ProverQA benchmark. Specifically, the model's performance increases by 21.4%, 44.0%, and 34.6% in the easy, medium, and hard subsets of the ProverQA benchmark, respectively. Similarly, fine-tuning on ProofWriter and FOLIO leads to performance gains on their corresponding test sets, as expected.

**Out-of-Distribution (OOD) Performance**  While the ID performance of finetuned models can exhibit the positive effects of specialized finetuning, OOD performance provides more insights as it demonstrates the generalizability of the finetuned models. As shown in Table 4, finetuning on

Table 4: Performance of Llama3.1-8B-Instruct finetuned with different FOL training sets. Green cells indicate in-distribution (ID) evaluation results, while red cells represent out-of-distribution (OOD) evaluation results. The change of performance after finetuning is shown in small fonts along with the evaluation results. †: the average *change* of performance across all OOD evaluation sets for each training configuration.

| FOL Training Set | ProverQA | | | ProntoQA | ProofWriter | FOLIO | OOD Avg $\Delta^\dagger$ |
|---|---|---|---|---|---|---|---|
| | Easy | Medium | Hard | | | | |
| *Llama3.1-8B-Instruct* | | | | | | | |
| - | 75.60 | 46.60 | 33.60 | 79.60 | 56.83 | 63.57 | - |
| *Finetuned Llama3.1-8B-Instruct* | | | | | | | |
| ProofWriter | 44.60 $^{-31.0}$ | 55.00 $^{+8.4}$ | 47.20 $^{+13.6}$ | **92.00** $^{+12.4}$ | **71.67** $^{+14.8}$ | 50.00 $^{-13.6}$ | -2.0 |
| FOLIO | 53.20 $^{-22.4}$ | 44.80 $^{-1.8}$ | 31.00 $^{-2.6}$ | 63.40 $^{-16.2}$ | 42.83 $^{-14.0}$ | **70.00** $^{+6.4}$ | -11.4 |
| ProverQA | **97.00** $^{+21.4}$ | **90.60** $^{+44.0}$ | **68.20** $^{+34.6}$ | 88.40 $^{+8.8}$ | 65.67 $^{+8.8}$ | 68.57 $^{+5.0}$ | **+7.5** |

ProverQA yields a performance increase on OOD test sets (ProntoQA, ProofWriter, FOLIO) by about 5% to 8.8%. Finetuning on ProverQA obtains the best OOD improvement on average with a large leading margin compared to models finetuned on ProofWriter and FOLIO, indicating the limited utility of these prior datasets for promoting the reasoning capabilities of LLMs. The model finetuned on ProofWriter achieves the highest performance on ProntoQA. This anomaly could be attributed to the high similarity between ProntoQA and ProofWriter: Both datasets utilize predefined templates for generating logic and corresponding natural language.

## 5.2 CASE STUDY

We show examples from the FOLIO test set in Table 5 to demonstrate how finetuning on ProverQA improves model performance. Before finetuning, models fail to solve the problems because of various reasons, such as using a common sense shortcut, being unfaithful to the context, and being unable to extend the reasoning chain. After training on ProverQA, models are able to derive a valid reasoning chain up to the goal based on the context, and the reasoning process rigidly adheres to the facts and rules.

## 6 CONCLUSIONS

In this work, we presented a novel framework called ProverGen for automatically generating FOL reasoning datasets by leveraging the strengths of LLMs in producing diverse natural language expressions and the rigor of symbolic provers in ensuring logical consistency. Using this framework, we developed ProverQA, a high-quality FOL reasoning benchmark that is both extensible and diverse. ProverQA encompasses complex rules, diverse natural language expressions, and faithful intermediate reasoning chains, providing a comprehensive evaluation benchmark for FOL reasoning. Our evaluation shows that SOTA LLMs struggle to solve ProverQA problems, even with CoT prompting, demonstrating the unique challenge posed by our dataset. We also finetuned Llama3.1-8B-Instruct on a separately generated training set and observed consistent performance improvements on both in-distribution and out-of-distribution test sets. We hope that our generation framework and the resulting dataset could contribute to the development of more robust and capable models in this area.

## 7 ETHICS STATEMENT

The keywords and names used in this paper are sourced entirely from public repositories and databases. Specifically, the names were collected from two public GitHub repositories under MIT

Table 5: Typical errors made by models and how finetuning correct them. Reasoning chains before and after finetuning are both displayed for comparison.

| Error Type | Context | Before Finetuning | After Finetuning |
|---|---|---|---|
| Common sense shortcut | All students are humans. If someone studies, then he/she is a student. A person either studies or teaches. No young person teaches. Jerry neither teaches nor is a manager. Rose is young or a student. Question: Jerry is not young. | A person either studies or teaches.No young person teaches. Therefore, a young person studies. If someone studies, then they are a student. All students are humans. Therefore, a young person is a human. People are either young or old. Therefore, a young person is not old. Jerry neither teaches nor is a manager. Therefore, Jerry studies. Since Jerry is a student and a human, Jerry is not old. Therefore, Jerry is young. Answer: false. | Jerry neither teaches nor is a manager. A person either studies or teaches. Jerry studies. Jerry studies. If someone studies, then he/she is a student. Jerry is a student. Jerry is a student. All students are humans. Jerry is a human. According to the context and the conclusions that have already been drawn, we can deduce that it is uncertain that Jerry is not young. Answer: uncertain. |
| Unfaithful to the context | Beijing is the capital of the People's Republic of China. Beijing is the world's most populous national capital city. Beijing is located in Northern China. Beijing hosted the 2008 Summer Olympics and 2008 Summer Paralympics Games. Beijing has hosted both the Summer and Winter Olympics, along with the Summer and Winter Paralympics. Question: Beijing has hosted both the 2008 Summer Olympics and a winter olympics. | Beijing hosted the 2008 Summer Olympics and 2008 Summer Paralympics Games. However, there is no information about Beijing hosting a Winter Olympics. Therefore, the statement is uncertain because it is not explicitly stated that Beijing has hosted a Winter Olympics, but it is also not explicitly stated that it has not. Answer: uncertain. | Beijing hosted the 2008 Summer Olympics and 2008 Summer Paralympics Games. Beijing has hosted both the Summer and Winter Olympics, along with the Summer and Winter Paralympics. Beijing has hosted both the 2008 Summer Olympics and a winter olympics. Therefore, it is true that Beijing has hosted both the 2008 Summer Olympics and a winter olympics. Answer: true. |
| Unable to extend the reasoning chain | Robert Lewandowski is a striker. Strikers are soccer players. Robert Lewandowski left Bayern Munchen. If a player left a team they no longer play for that team. Question: Robert Lewandowski plays for Bayern Munchen. | Robert Lewandowski is a striker. Strikers are soccer players. Therefore, Robert Lewandowski is a soccer player. Robert Lewandowski left Bayern Munchen. If a player left a team, they no longer play for that team. Therefore, Robert Lewandowski no longer plays for Bayern Munchen. However, the question is asking if he plays for Bayern Munchen, not if he left. The information does not state that he currently plays for another team. Answer: uncertain. | Robert Lewandowski left Bayern Munchen. If a player left a team they no longer play for that team. Robert Lewandowski does not play for Bayern Munchen. Therefore, it is false that Robert Lewandowski plays for Bayern Munchen. Answer: false. |

licenses, and the keywords were obtained from WordNet. All sources have licenses permitting free usage, and our work complies with these terms. Our dataset does not involve human participants, nor does it present any immediately harmful insights. For the release, we plan to open source the code for creating the benchmark and the dataset we created for the experiments in this paper.

## 8 REPRODUCIBILITY STATEMENT

For our submission, we have uploaded the entirety of the source code as a zipped file that has been properly anonymized. The source code contains inline documentation that details purpose and usage of different parts of the codebase. In addition, we also include the full set of model's responses and the evaluation script. As discussed in the ethics statement, we plan to more formally release ProverQA to the public as an open source repository with thorough details that describes the framewrok, outlines the code, and details its usage.

## ACKNOWLEDGMENTS

This work is supported by National Key Research and Development Program of China (Grant No. 2024YFB3311902), National Natural Science Foundation of China (Grant No. 62173017), Beijing Natural Science Foundation (Grant No. L241018), Beijing Nova Program (Grant No. 2024085), and Shanghai Artificial Intelligence Laboratory.

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

## A  PROMPT FOR GENERATING BACKGROUND STORY

```
System:
You will be given a keyword and a name (can be a person's name or an animals' name). Your task
is to generate a background story with no more than 150 words according to the keyword about
this name. Your answer should be in JSON format with keys: category, story.

User:
keyword: swift
name: Jack

Assistant:
{
  "category": "cat",
  "story": "Jack was a sleek, agile cat known for his swift movements. Born in the bustling
            city, he quickly learned to navigate the narrow alleyways and rooftops with ease.
            His speed and agility made him a legend among the local feline community. Jack's
            favorite pastime was chasing the elusive city birds, darting through the air with
            unmatched precision. Despite his wild spirit, he had a soft spot for the elderly
            woman who fed him daily. She named him Jack after the nimble character from her
            favorite childhood story, believing that his swiftness was a sign of good fortune.
            Jack's swift nature wasn't just physical; he had a quick wit and an uncanny ability
            to sense danger, keeping him safe in the urban jungle. His legend grew, and he
            became a symbol of agility and cleverness in the neighborhood."
}

User:
keyword: success
name: Amy
```

```
Assistant:
{
  "category": "human",
  "story": "Amy grew up in a small town where opportunities were scarce. Despite the odds, she
            was determined to make something of herself. After excelling in high school, she
            earned a scholarship to a prestigious university. There, she majored in computer
            science and quickly became known for her innovative ideas. Upon graduation, Amy co-
            founded a tech startup that focused on sustainable technology. Her company quickly
            gained traction, securing major investments and partnerships. Amy's relentless
            work ethic and visionary leadership turned her startup into a multi-million dollar
            enterprise within five years. Today, she is celebrated not only for her business
            acumen but also for her commitment to mentoring young entrepreneurs and giving back
            to her community. Amy's story is a testament to the power of perseverance, vision,
            and hard work."
}

User:
keyword: loyal
name: Bella

Assistant:
{
  "category": "dog",
  "story": "Bella was a golden retriever with a heart as pure as gold. From the moment she was
            adopted by the Thompson family, she became their steadfast guardian and companion.
            Bella had an uncanny ability to sense when someone in the family needed comfort,
            always there with a nuzzle or a wagging tail. Her loyalty was most evident when
            young Timmy got lost in the woods behind their house. Bella tracked his scent
            tirelessly, refusing to rest until she found him. She stayed by his side through
            the cold night, keeping him warm and safe until help arrived. Bella's unwavering
            loyalty and love made her not just a pet, but a cherished member of the
            family."
}

User:
keyword: [KEYWORD]
name: [NAME]
```

# B  PROMPTS FOR TRANSLATING LOGIC EXPRESSIONS INTO NATURAL LANGUAGE

We utilized three examples for each type of logical expression to guide the LLMs in translating them into natural language. An example of the prompt we used in our experiments is shown below.

```
System:
You will be provided a logic expression, a reference of the existed predicate in the logic
expression, and a background story. Your task is to replace the placeholders in the logic
expression with appropriate predicates (no more than 5 words) so that the provided logic
expression represent a real world common sense rule.

Do not use 'not'. Do not use the words in the forbidden list. The words that are similar
in meaning to the words in the forbidden list are also not allowed.

Your answer is not required to closely connected to the background story. You can use any
predicates you like as long as their length is less than 5.

Your answer should be in JSON format with the provided keys.

User:
background story:
Moriarty was a large, fluffy Maine Coon cat with a surprisingly warm and gentle nature.
Despite his intimidating size and the mysterious name inspired by a famous literary villain,
Moriarty was the epitome of warmth and comfort. He lived in a cozy bookstore run by Mrs.
Lindon, an elderly woman who had taken him in as a stray. Moriarty had a special talent for
sensing when a customer needed a bit of extra warmth, often curling up in their laps as they
read. His thick fur and gentle purring provided solace to many, especially during the cold
winter months. Moriarty's presence turned the bookstore into a beloved local haven, where
people came not just for the books, but for the warm, comforting presence of their favorite
feline friend.

reference: is_fluffy
is_fluffy(Moriarty) → (warm_and_gentle(Moriarty) ⊕ is_calming(Moriarty)): If Moriarty is
fluffy, then he is either warm and gentle or calming, but not both.
forbidden list: ['warm_and_gentle', 'provide_comfort', 'attract_customers', 'is_friendly',
'brings_joy', 'is_fluffy', 'is_calming']
```

```
logic expression: ∀x (F7(x) → (is_fluffy(x) ∧ F8(x)))
keys: ['F7', 'F8', 'universal_rule', 'specific_rule']
Note: x belongs to cat.

Assistant:
{
  "F7": "playful",
  "F8": "love_attention",
  "universal_rule": "All playful cats are fluffy and love attention.",
  "specific_rule": "If Moriarty is playful, then he is both fluffy and loves attention."
}

---------------------
Another Two Examples
---------------------

User:
background story:
Colt was a brilliant botanist with a passion for rare plant species. He spent years studying
the Avicenniaceae family, specifically the mangrove plants, which thrive in the harshest of
coastal environments. Colt's groundbreaking research on these unique plants led to the
development of innovative solutions for coastal conservation. He discovered that the Avicennia
species possessed remarkable adaptability, allowing them to survive in areas with high salinity
and limited oxygen. Colt's findings opened doors to new methods for restoring damaged
ecosystems and protecting endangered species. As a leading expert in his field, Colt
collaborated with international organizations to implement his research, helping to preserve
the delicate balance of our planet's mangrove ecosystems. His dedication to the Avicenniaceae
family left a lasting impact on the scientific community.

reference: contribute_to_environmental_conservation
dedicated_to_research(Colt) → (make_groundbreaking_discoveries(Colt) ∧
contribute_to_environmental_conservation(Colt)): If Colt is dedicated to research, then he
will make groundbreaking discoveries and contribute to environmental conservation.
forbidden list: ['dedicated_to_research', 'make_groundbreaking_discoveries',
'contribute_to_environmental_conservation', 'pursue_higher_education', 'become_leading_expert']

logic expression: ∀x (F5(x) → (contribute_to_environmental_conservation(x) ∧ F6(x)))
keys: ['F5', 'F6', 'universal_rule', 'specific_rule']
Note: x belongs to human.
```

## C  POTENTIAL SHORTCUT PROBLEM

To further investigate the potential use of shortcuts, we conducted an experiment by removing universal rules from 60 randomly selected instances in ProverQA. We evaluated GPT-4 and Llama-3.1-70B-Instruct on this *corrupted* dataset. If the models were relying heavily on inherent knowledge as shortcuts, their performance would remain roughly unaffected despite the absence of universal rules. However, the results in Table 6 showed a significant drop in performance, indicating that these models do not heavily rely on background knowledge to solve the problems.

Table 6: Comparative performance of GPT-4o and Llama-3.1-70B-Instruct on the original dataset and the corresponding corrupted version.

| Model | Prompting Strategy | Original | Corrupted | Acc Δ |
|---|---|---|---|---|
| GPT-4o | Direct | 58.33 | 43.33 | -15.00 |
| GPT-4o | CoT | 68.33 | 45.00 | -23.33 |
| Llama-3.1-70B-Instruct | Direct | 65.00 | 48.33 | -16.67 |
| Llama-3.1-70B-Instruct | CoT | 65.00 | 53.33 | -11.67 |

## D  EXAMPLES FROM THE THREE SUBSETS OF PROVERQA

**ProverQA-Easy**

```
Context:
For every elephant, the elephant is either domesticated or wild, but not both.
Anakin is a domesticated elephant.
```

```
Question: Based on the above information, is the following statement true, false, or uncertain?
Anakin is not a wild elephant.

Options:
A) True
B) False
C) Uncertain

The correct option is: A

------------------------------------------Proofs------------------------------------------

------------------Step1------------------

fact1: Anakin is a domesticated elephant.
rule: For every elephant, the elephant is either domesticated or wild, but not both.
conclusion: Anakin is not a wild elephant.

Therefore, it is true that Anakin is not a wild elephant. The correct option is: A.
```

## ProverQA-Medium

```
Context:
Everyone who either practices waltz or masters tango (but not both) has good dance skills.
Sawyer has charisma. Sawyer is a performer. If Sawyer has good dance skills and has charisma,
then he is a successful dancer. Sawyer is either a musician or a poet, but not both. Alice is
a poet. Sawyer is a poet. If Sawyer is a performer, then he is either skilled at dancing or a
musician, but not both.

Question: Based on the above information, is the following statement true, false, or uncertain?
Sawyer is a successful dancer.

Options:
A) True
B) False
C) Uncertain

The correct option is: A

------------------------------------------Proofs------------------------------------------

------------------Step1------------------

fact1: Sawyer is a poet.
rule: Sawyer is either a musician or a poet, but not both.
conclusion: Sawyer is not a musician.

------------------Step2------------------

fact1: Sawyer is a performer.
fact2: Sawyer is not a musician.
rule: If Sawyer is a performer, then he is either skilled at dancing or a musician, but
not both.
conclusion: Sawyer has good dance skills.

------------------Step3------------------

fact1: Sawyer has good dance skills.
fact2: Sawyer has charisma.
rule: If Sawyer has good dance skills and has charisma, then he is a successful dancer.
conclusion: Sawyer is a successful dancer.

Therefore, it is true that Sawyer is a successful dancer. The correct option is: A.
```

## ProverQA-Hard

```
Context:
Buster does not respond to calls. All dogs that listen to commands and behave properly are
well-trained. Buster is either obedient or playful, but not necessarily both. Buster follows
routines. Buster is either playful or loving (or both). If Buster either responds to calls or
follows routines (but not both), then he listens to commands. All well-trained dogs are
obedient and have good manners. Buster behaves properly.

Question: Based on the above information, is the following statement true, false, or uncertain?
Buster is loving.

Options:
A) True
```

```
B) False
C) Uncertain

The correct option is: A

-------------------------------------------Proofs---------------------------------------------

------------------Step1------------------

fact1: Buster does not respond to calls.
fact2: Buster follows routines.
rule: If Buster either responds to calls or follows routines (but not both), then he listens
to commands.
conclusion: Buster listens to commands.

------------------Step2------------------

fact1: Buster listens to commands
fact2: Buster behaves properly.
rule: All dogs that listen to commands and behave properly are well-trained.
conclusion: Buster is well-trained.

------------------Step3------------------

fact1: Buster is well-trained.
rule: All well-trained dogs are obedient and have good manners.
conclusion: Buster is obedient.

------------------Step4------------------

fact1: Buster is obedient.
rule: Buster is either obedient or playful, but not necessarily both.
conclusion: Buster is not playful.

------------------Step5------------------

fact1: Buster is not playful.
rule: Buster is either playful or loving (or both).
conclusion: Buster is loving.

Therefore, it is true that Buster is loving. The correct option is: A.
```

# E    EXAMPLES OF THE TWO PROMPTING METHODS

The prompts of the two prompting strategies we used in our experiments are listed below. We adopt 2-shot in-context learning for both `Standard` and `CoT` prompting strategies.

**Standard Prompting**

```
System:
Given a problem statement as contexts, the task is to answer a logical reasoning question.
Your answer should be in JSON format with key: answer.

Context:
Anyone who can solve problems and communicate effectively has practical skills. If Paola
appreciates beauty, then she values precision and enjoys manual work. Paola appreciates beauty.
Paola is either skilled in mathematics or has practical skills, but not both. If someone enjoys
manual work and values attention to detail, then they can craft with their hands. Paola is
either skilled in mathematics or artistically inclined, but surprisingly, she is actually both.
Paola values attention to detail. Anyone who crafts with their hands has practical skills.

Question: Based on the above information, is the following statement true, false, or uncertain?
Paola is artistically inclined.

Options:
A) True
B) False
C) Uncertain

The correct option is: {
  "answer": "A"
}
------------------
Another Example
------------------
Context:
[[CONTEXT]]
```

```
Question: [[QUESTION]]

Options:
[[OPTIONS]]

The correct option is:
```

### CoT Prompting

```
System:
Given a problem statement as contexts, the task is to answer a logical reasoning question.
Your answer should be in JSON format with keys: reasoning, answer.

Context:
Anyone who can solve problems and communicate effectively has practical skills. If Paola
appreciates beauty, then she values precision and enjoys manual work. Paola appreciates beauty.
Paola is either skilled in mathematics or has practical skills, but not both. If someone enjoys
manual work and values attention to detail, then they can craft with their hands. Paola is
either skilled in mathematics or artistically inclined, but surprisingly, she is actually both.
Paola values attention to detail. Anyone who crafts with their hands has practical skills.

Question: Based on the above information, is the following statement true, false, or uncertain?
Paola is artistically inclined.

Options:
A) True
B) False
C) Uncertain

The correct option is: {
  "reasoning": "Paola appreciates beauty. If Paola appreciates beauty, then she values
               precision and enjoys manual work. Paola enjoys manual
               work. Paola values attention to detail. If someone enjoys manual work and
               values attention to detail, then they can craft with their hands. Paola crafts
               with her hands. Paola crafts with her hands. Anyone who crafts with their hands
               has practical skills. Paola has practical skills. Paola has practical skills.
               Paola is either skilled in mathematics or has practical skills, but not both.
               Paola is not skilled in mathematics. Paola is not skilled in mathematics. Paola
               is either skilled in mathematics or artistically inclined, but surprisingly,
               she is actually both. Paola is artistically inclined. Therefore, it is true
               that Paola is artistically inclined. The correct option is: A.",
  "answer": "A"
}
------------------
Another Example
------------------
Context:
[[CONTEXT]]

Question: [[QUESTION]]

Options:
[[OPTIONS]]

The correct option is:
```

## F  TRAINING DETAILS

To create the ProverQA train set, we started by generating 5,000 instances for each difficulty level: easy, medium, and hard. We then applied data augmentation techniques to expand the dataset. Specifically, for each problem, we broke down every step of its reasoning chain to form new questions. Additionally, we replaced the final conclusions with random facts to create problems whose answer is uncertain. This process expanded our generated data pool to 50,000 instances.

Our experiments show that, as the number of training instances increased, the model's performance consistently improved on in-distribution datasets and two out-of-distribution datasets: ProofWriter and ProntoQA. However, performance on FOLIO fluctuated as the number of training instances increases. After further experimentation, we found that 5,000 is a sweet spot that reaches a good balance between in-distribution and out-of-distribution performance. Therefore, we sample 5,000 instances from the generated data pool as the final training set for finetuning experiments.

Since FOLIO lacks reasoning chains, we could not apply the same augmentation techniques on it. We used the full set of FOLIO, consisting of 1,001 instances. For ProofWriter, which includes multiple questions per context similar to our data augmentation, we did not apply further augmentation.

We also finetuned the model on the full set of ProofWriter, which has 45,290 instances. The performance is comparable compared to the performance of model trained on a sampled subset consisting of 5,000 instances, indicating that the model may already be saturated at this size for an 8B parameter model. As a result, we report the results of the sampled subset.

For all finetuning experiments, we experimented with different training hyper-parameters, such as training epochs, learning rate and training set size, and selected the best configurations on held-out validation sets.

## G   EXPERIMENTING WITH 5-SHOT EXAMPLES

To further explore the impact of few-shot examples, we conducted additional experiments using 5-shot prompts. The results are presented in Table 7. It can be concluded that incorporating more examples (5-shot vs. 2-shot) into the prompt did not consistently improve the performance of LLMs. This inconsistency may stem from several factors, including variations in the models' in-context learning capabilities.

## H   QUALITY CONTROL

The ProverGen framework employs three heuristic methods to ensure the quality of the generated data, addressing different aspects of data integrity and consistency.

- **Logic Validation**: Our framework includes a robust logic validation step. For each instance, we input both core premises and distractions into the symbolic prover to ensure they correctly deduce the conclusion's truth value.
- **Conflicts Resolution**: During rule translation, we check for previously used predicates to avoid redundancy and potential conficting facts. Additionally, we utilize LLMs to assess whether the generated universal rules align with real-world knowledge. In cases of conflict, we opt for specific rules instead.
- **Translation Quality Control:** We apply a heuristic method to ensure that all involved entities appear in both symbolic expression and natural language expression. For example, when translating "poet(Sawyer)" into "Sawyer is a poet", we verify that both the name "Sawyer" and the predicate "poet" are present in the translation.

Additionally, the training part of the paper also serves as a quality-checking process. Finetuning on the generated dataset enhances the performance of LLMs on both in-distribution and Out-of-Distribution datasets, indicating the relatively high quality of the generated data.

To further investigate the quality of the generated data, we performed a manual evaluation of 60 randomly sampled examples (20 from each subset) from our dataset. The evaluation revealed no translation errors or indirect contradictions, suggesting that the likelihood of potential errors is less than 2%.

Table 7: The performance of LLMs in the 5-shot setting. The values in parentheses indicate the change in performance relative to the 2-shot setting.

| | ProverQA | | | ProntoQA | ProofWriter | FOLIO | Avg △ |
|---|---|---|---|---|---|---|---|
| | Easy | Medium | Hard | | | | |
| *Standard Prompting* | | | | | | | |
| GPT-4o | 85.20 | 68.40 | 43.80 | 88.20 | 57.33 | 72.86 | -0.37 |
| | (-2.00) | (-0.20) | (-2.40) | (-3.60) | (+1.00) | (+5.00) | |
| Claude-3.5-Sonnet | 93.80 | 76.80 | 51.00 | 85.20 | 50.33 | 74.29 | +2.23 |
| | (+8.20) | (+8.60) | (+8.20) | (-3.40) | (-4.67) | (-3.56) | |
| Llama3.1-8B-Instruct | 64.40 | 41.80 | 38.00 | 58.00 | 44.50 | 46.43 | +2.67 |
| | (+17.80) | (-1.20) | (-1.00) | (+7.60) | (+0.70) | (-7.86) | |
| Llama3.1-70B-Instruct | 82.80 | 64.40 | 49.20 | 74.00 | 51.33 | 69.29 | -0.26 |
| | (+0.80) | (+0.20) | (+1.60) | (-6.60) | (+1.00) | (+1.43) | |
| Mistral-7B-Instruct | 57.40 | 48.40 | 37.40 | 52.40 | 47.83 | 52.86 | +1.48 |
| | (+0.60) | (+1.60) | (+0.20) | (+2.40) | (+5.50) | (-1.43) | |
| Mistral-Large-Instruct | 85.00 | 71.40 | 53.60 | 67.40 | 63.50 | 76.43 | +0.91 |
| | (+0.40) | (+2.20) | (+4.00) | (-3.60) | (+3.17) | (-0.71) | |
| Mixtral-8x22B-Instruct | 77.60 | 58.00 | 39.60 | 60.00 | 40.50 | 73.57 | -0.37 |
| | (+2.20) | (+0.60) | (+0.60) | (-5.20) | (+0.33) | (-0.72) | |
| *CoT Prompting* | | | | | | | |
| GPT-4o | 95.00 | 81.20 | 55.00 | 99.20 | 71.00 | 74.29 | +2.10 |
| | (+0.80) | (+1.80) | (+5.00) | (-0.80) | (+3.67) | (+2.15) | |
| Claude-3.5-Sonnet | 92.00 | 81.60 | 61.00 | 95.80 | 75.50 | 82.86 | -0.45 |
| | (-3.20) | (-2.00) | (+4.60) | (-3.40) | (-0.83) | (+2.15) | |
| Llama3.1-8B-Instruct | 80.60 | 45.40 | 31.20 | 74.80 | 58.67 | 57.86 | -1.21 |
| | (+5.00) | (-1.20) | (-2.40) | (-4.80) | (+1.84) | (-5.68) | |
| Llama3.1-70B-Instruct | 92.60 | 73.40 | 50.20 | 92.80 | 67.00 | 75.00 | -0.04 |
| | (+2.20) | (+0.20) | (+3.40) | (-2.60) | (-4.17) | (+0.71) | |
| Mistral-7B-Instruct | 65.60 | 50.20 | 38.00 | 65.80 | 48.33 | 63.57 | -0.68 |
| | (-6.40) | (-0.80) | (-3.80) | (+4.60) | (+2.33) | (-0.01) | |
| Mistral-Large-Instruct | 94.40 | 77.60 | 56.60 | 99.00 | 77.67 | 82.86 | +1.98 |
| | (+1.80) | (+1.80) | (+4.40) | (+0.40) | (+4.17) | (-0.71) | |
| Mixtral-8x22B-Instruct | 91.00 | 73.80 | 50.20 | 86.20 | 59.67 | 72.14 | +3.36 |
| | (+3.40) | (+7.00) | (+2.60) | (+6.60) | (+2.00) | (-1.43) | |

