# OpenReview forum: "Large Language Models Meet Symbolic Provers for Logical Reasoning Evaluation"
_ICLR.cc/2025/Conference — ICLR 2025 Poster_

### Official Review · Reviewer_oQpi · 2024-10-26

**Soundness:** 1
**Presentation:** 3
**Contribution:** 1
**Rating:** 3
**Confidence:** 4

**Summary:**

This paper introduces ProverGen, a synthetically generated benchmark for deductive reasoning evaluation, which consists of 1.5K examples. ProverGen resembles prior work such as ProntoQA, where it generates a ground-truth reasoning path and then converts it into a deductive reasoning problem. On top of this, the authors use LLM to translate the rules and facts into NL statements based on a LLM generated background facts, making the problem more natural.

**Strengths:**

see below

**Weaknesses:**

## Novelty

As mentioned above, the creation of this dataset largely resembles that of ProntoQA. The main difference is the LLM translated facts and rules that make the statements semantically more natural and diverse. However, it is not sufficiently demonstrated how this aspect contributes to the novelty of this benchmark. It is unclear what limitations or insights of LLMs are revealed by this dataset that others could not. This, together with the quality and significance issues to be discussed below, significantly undermine the novelty of this work.


## Quality

There are several issues with the creation of the dataset.

Some FOL reasoning cases are potentially not covered by this benchmark:
- Judging from the pipeline and examples, the "goal" fact is an instantiated literal with unary predicate such as *Elegant(Sawyer)*, and *Swift(jack)*, but it does not cover cases with binary predicates or those with higher-order of arity, such as *FriendOf(Amy, Bob)*. It also does not cover cases of composite facts, such as *FriendOf(Amy, Bob) \land Person(Bob)*

Lack of dataset quality check:
- There lacks a check on whether the translated NL statement actually align with the ground-truth FOL rule
- Furthermore, the translated universal rule is only checked by an LLM on whether it aligns with commonsense. This could be noisy as the LLM can hallucinate.
- Without a quantitative measure on the quality of the translation, it is difficult to assess the dataset quality. At least one should provide the accuracy of the translation on a small held-out set that has manual annotations.

Finetuning results interpretation:
- As mentioned in the appendix, model finetuned on FOLIO is trained with only 1K examples, while the other two has 50K. I'm not sure if one can reliably draw any conclusion by comparing it with the other two as the training data difference is too big.

## Clarity

This paper is generally easy to follow.

## Significance

Another major concern of mine is the significance of this benchmark.
- This dataset contains only 1.5K examples with synthetically generated reasoning chains and NL translation that has no direct verification. While the authors generated a training set for experiments in section 5, it was not presented as part of the contribution and there also lacks direct verification on its quality and alignment. That said, this dataset really can only be used to evaluate an LLM on a specific reasoning task, i.e., deductive reasoning.
- Nevertheless, an evaluation dataset could also be a concrete contribution, but in order to be significant, it needs to reveal something new that was neglected and overlooked in other benchmarks, or bring new or significant harder challenges to the table. Unfortunately, this is not sufficiently demonstrated in the paper. Throughout the comparison in Table 1, ProverGen is on par with other benchmarks with only the hard set being somewhat more challenging. But where is the insight? Does the model fail because of an important property that is exclusively tested in ProverGen is missing? What can people learn from ProverGen about the LLM's reasoning capability that benchmarks fail to reveal?

That said, the authors need to address this issue before this work can be considered significant.

**Questions:**

see above

---

> ### Author Response · Authors · 2024-11-21
> **Response to Reviewer oQpi (1/2)**
>
> Thank you for your comments and constructive suggestions. We address specific weaknesses and questions below.
>
> > W1: As mentioned above, the creation of this dataset largely resembles that of ProntoQA. The main difference is the LLM translated facts and rules that make the statements semantically more natural and diverse. However, it is not sufficiently demonstrated how this aspect contributes to the novelty of this benchmark.
>
> Thank you for your feedback. We appreciate the opportunity to clarify the distinctions of our dataset compared to ProntoQA, beyond the use of LLM translation.
> 1. **Generation Process**: Our framework employs a top-down approach to generate logic skeletons, contrasting with ProntoQA’s reliance on predefined logic templates. This method avoids the risk of generating disconnected or contradictory premises, ensuring more coherent datasets.
> 2. **Symbolic Prover Integration**: Our framework includes a symbolic prover, allowing for comprehensive coverage of possible logic expressions and the generation of diverse reasoning trees. In contrast, ProntoQA's approach is limited to predefined reasoning paths, which restricts the diversity of reasoning trees.
> 3. **Complexity Control**: Our framework offers extensive customization of reasoning steps and complexity, which is not available in previous datasets including ProntoQA.
> We believe these aspects collectively contribute to the novelty and significance of our work. We are pleased to note that these contributions have been recognized by the other three reviewers.
>
> > W2: It is unclear what limitations or insights of LLMs are revealed by this dataset that others could not. This, together with the quality and significance issues to be discussed below, significantly undermine the novelty of this work.
>
> Thank you for your feedback. Our work highlights the limitations of SOTA models in handling FOL problems requiring long reasoning chains and complex rules, as evidenced in Table 2. Unlike existing datasets, ours introduces specific distractions that significantly impair LLMs' reasoning capabilities (see Table 3). These insights contribute to a deeper understanding of LLM limitations and emphasize the dataset's novelty.
>
> > W3: Some FOL reasoning cases are potentially not covered by this benchmark:Judging from the pipeline and examples, the "goal" fact is an instantiated literal with unary predicate such as Elegant(Sawyer), and Swift(jack), but it does not cover cases with binary predicates or those with higher-order of arity, such as FriendOf(Amy, Bob). It also does not cover cases of composite facts, such as FriendOf(Amy, Bob) \land Person(Bob)
>
> We apologize for any confusion. The "goal" in our framework indeed accommodates both facts and rules, as noted in Section 3.1: "The goal of FOL reasoning is to determine whether a given goal G (which can be a fact or a rule) is ...". This encompasses unary, binary, and higher-order predicates, as well as composite facts like FriendOf(Amy, Bob) \land Person(Bob).
>
> > W4: Lack of dataset quality check:
> > - There lacks a check on whether the translated NL statement actually align with the ground-truth FOL rule
> > - Furthermore, the translated universal rule is only checked by an LLM on whether it aligns with commonsense. This could be noisy as the LLM can hallucinate.
> > - Without a quantitative measure on the quality of the translation, it is difficult to assess the dataset quality. At least one should provide the accuracy of the translation on a small held-out set that has manual annotations.
>
> Thank you for your suggestion. We included quality control in our translation process, as detailed in our response to W4 for Reviewer 64s6. We manually verified translations on 60 instances sampled from our benchmark and found no errors, demonstrating the effectiveness of our quality control measures.
>
> Additionally, the training part of the paper also serves as quality checking. Finetuning on the generated dataset enhances the performance of LLMs on both in-distribution and OOD datasets, indicating the relatively high quality of the generated data.
>
> Despite the above quality control processes and finetuning experiments, we agree that our framework can be further improved by introducing more advanced quality control processes. We will explore more about it in our future work.

---

> ### Author Response · Authors · 2024-11-21
> **Response to Reviewer oQpi (2/2)**
>
> > W5: Finetuning results interpretation: As mentioned in the appendix, model finetuned on FOLIO is trained with only 1K examples, while the other two has 50K. I'm not sure if one can reliably draw any conclusion by comparing it with the other two as the training data difference is too big.
>
> The finetuning results for ProverGen and ProofWriter were indeed conducted on 5k instances, not 50k. The training set of FOLIO only has 1k instances, which is why it was trained with only 1k examples. These numbers are clarified in Appendix E.
>
> To ensure fairness in comparison, we optimized training configurations through various hyperparameter experiments, including epochs, learning rates, and data sizes, based on performance on validation sets. Specifically, the FOLIO models were trained for 5 epochs, while the other datasets were trained for only 1 epoch, allowing us to balance the differences in data availability.
>
> > W6: This dataset contains only 1.5K examples with synthetically generated reasoning chains and NL translation that has no direct verification. While the authors generated a training set for experiments in section 5, it was not presented as part of the contribution and there also lacks direct verification on its quality and alignment. That said, this dataset really can only be used to evaluate an LLM on a specific reasoning task, i.e., deductive reasoning.
>
> Thank you for your feedback. Our framework is designed to be scalable, allowing for the generation of additional examples as required. While we did not initially highlight the dataset as a primary contribution, its utility extends beyond mere evaluation. In Section 5, we demonstrate its effectiveness in enhancing LLMs' first-order logic reasoning abilities during training. Additionally, the value of our dataset has been recognized by other reviewers, reinforcing its potential impact.
>
> > W7: Nevertheless, an evaluation dataset could also be a concrete contribution, but in order to be significant, it needs to reveal something new that was neglected and overlooked in other benchmarks, or bring new or significant harder challenges to the table. Unfortunately, this is not sufficiently demonstrated in the paper. Throughout the comparison in Table 1, ProverGen is on par with other benchmarks with only the hard set being somewhat more challenging. But where is the insight? Does the model fail because of an important property that is exclusively tested in ProverGen is missing? What can people learn from ProverGen about the LLM's reasoning capability that benchmarks fail to reveal?
>
> Here are the key insights provided in our work:
> 1. Our study shows that LLMs still struggle significantly with complex reasoning tasks, particularly those involving long chains of reasoning and intricate logic.
> 2. We conducted ablation studies revealing that distracting factors and shuffled premises notably impact model accuracy, an area not previously explored in existing benchmarks.
> 3. Our scalable, complex, natural, and diverse FOL dataset enhances LLMs' logical reasoning capabilities , even on out-of-distribution datasets.
>
> Additionally, other reviewers have recognized the value of our dataset, acknowledging its role in presenting new challenges to current LLMs and offering fresh insights into improving logical reasoning skills.
>
> We greatly appreciate the reviewer's feedback and welcome further discussions.

---

> > ### Comment · Reviewer_oQpi · 2024-12-02
> > **Response**
> >
> > Thanks for the response
> >
> > > Generation Process: Our framework employs a top-down approach to generate logic skeletons, contrasting with ProntoQA’s reliance on predefined logic templates. This method avoids the risk of generating disconnected or contradictory premises, ensuring more coherent datasets.
> >
> > Prior work such as ProntoQA and RuleTaker all come with source code for generation and verification, while others like BoardGameQA, Proofwriter contained chains built in various ways. It is not sufficiently demonstrated how more novel is the top-down approach and how more coherent is the dataset compared to the previous ones.
> >
> > > Symbolic Prover Integration: Our framework includes a symbolic prover, allowing for comprehensive coverage of possible logic expressions and the generation of diverse reasoning trees. In contrast, ProntoQA's approach is limited to predefined reasoning paths, which restricts the diversity of reasoning trees.
> >
> > Deductive reasoning benchmarks all share the same tree-like reasoning chain with some modifications, for example, BoardGameQA also introduces distractions and contradictory facts. To justify this, one needs a clear definition of *diversity* and quantitative compare it against others
> >
> > > Complexity Control: Our framework offers extensive customization of reasoning steps and complexity, which is not available in previous datasets including ProntoQA. We believe these aspects collectively contribute to the novelty and significance of our work. We are pleased to note that these contributions have been recognized by the other three reviewers.
> >
> > As mentioned above, work such as ProntoQA and RuleTaker all come with source code for generation, where one can also customize for complexity.
> >
> > > W3 W4
> >
> > My concerns are addressed
> >
> > > To ensure fairness in comparison, we optimized training configurations through various hyperparameter experiments
> >
> > Comparing models trained on 1K data versus 5k does not make any sense even if the former is trained for 5 more times. I do not think one can reasonably draw any conclusion from this comparison.
> >
> > > Here are the key insights provided in our work:
> > > Our study shows that LLMs still struggle significantly with complex reasoning tasks, particularly those involving long chains of reasoning and intricate logic.
> >
> > This is also sufficiently demonstrated in prior work such as ProntoQA, RuleTaker, BoardGameQA, and Proofwriter.
> >
> > > We conducted ablation studies revealing that distracting factors and shuffled premises notably impact model accuracy, an area not previously explored in existing benchmarks.
> >
> > The effect of distraction in deductive reasoning is also demonstrated in BoardGameQA
> >
> > > Our scalable, complex, natural, and diverse FOL dataset enhances LLMs' logical reasoning capabilities , even on out-of-distribution datasets.
> >
> > A verified training dataset could be a good contribution, though I'm not very confident that checking 60 samples would be sufficient to verify the dataset.
> >
> > ---
> >
> > In summary, my main concern about this work remains and some of the critical issues are left unaddressed
> > - It is still unclear what is unique about this dataset. While it seems to be created in a slightly different way than prior work, it is unclear what can people learn from ProverGen about the LLM's reasoning capability that benchmarks fail to reveal as I replied above.
> > - Also, my concern that "the LLM translated facts and rules that make the statements semantically more natural and diverse; it is not sufficiently demonstrated how this aspect contributes to the novelty of this benchmark" was not addressed. Concretely, while the background story was generated by LLM, it does not align with the chain rule tree generated other than providing a context for rule and fact translation. This dataset is effectively still synthetic as the reasoning chains are not drawn from real-world natural language distribution.
> >
> > I have read other reviewers' comments, and while I partially agree with their views, I still believe the issues I have are critical and potentially overlooked. That said, I'm keeping my score.

---

> ### Author Response · Authors · 2024-11-25
>
> Dear Reviewer oQpi,
>
> We sincerely appreciate the time and effort you have dedicated to reviewing our work!
>
> As the discussion period comes to an end, we would deeply appreciate it if you could take a moment to review our responses and let us know if they adequately address your concerns and questions.

---

> ### Author Response · Authors · 2024-12-03
> **Response to Reviewer oQpi (1/3)**
>
> > Follow-up Q1: Prior work such as ProntoQA and RuleTaker all come with source code for generation and verification, while others like BoardGameQA, Proofwriter contained chains built in various ways. It is not sufficiently demonstrated how more novel is the top-down approach and how more coherent is the dataset compared to the previous ones.
>
> While having source code is valuable for reproduction, the key innovation lies in how the data is generated and the inherent properties of the dataset. Our framework introduces several novel aspects:
>
> 1. **Complex Logic Coverage**: Our dataset incorporates a broader range of logical constructs, such as $\(A \to B\), \(A \oplus B\), \(A \lor B\), \(A \land B\), \(A \land (B \land C)\), \((A \land B) \to C\)$, and others. In contrast, datasets like ProntoQA and RuleTaker are limited to simpler logics such as $\(A \to B\)$ and basic conjunctions and implications.
>
> 2. **Coherent Language Use**: We have focused on ensuring greater linguistic coherence within our dataset. For example, our sentences are contextually linked and logically consistent, unlike some examples from other datasets, which may seem disjointed or arbitrary:
>     - ProntoQA: "Every tumpus is not angry. Tumpuses are rompuses."
>     - ProofWriter: "The cat eats the bear. The cat is green. The cat is kind."
>     - ProverGen: "If Sawyer is a performer, then he either has good dance skills or is a musician."
>
> 3. **Ease of Complexity Management**: Our framework allows for straightforward incorporation of new logical constructs by simply adding them to the rule pool. This flexibility contrasts with the need for creating new templates or retraining models in other datasets like ProntoQA, RuleTaker, and ProofWriter.
>
> Additionally, BoardGameQA is not an FOL dataset, making direct comparisons less relevant. Our dataset's focus on FOL ensures that it remains coherent and applicable to the specific domain of logical reasoning.
>
> > Follow-up Q2: Deductive reasoning benchmarks all share the same tree-like reasoning chain with some modifications, for example, BoardGameQA also introduces distractions and contradictory facts. To justify this, one needs a clear definition of diversity and quantitative compare it against others
>
> We would like to argue that the mere presence of tree-like reasoning chains does not inherently signify novelty. Rather, the innovation lies in the methods used to generate these chains and their structural characteristics. Moreover, it is important to note that not all deductive reasoning benchmarks utilize tree-like reasoning chains. For instance, the reasoning path in ProntoQA is explicitly linear, as stated in the original paper: "To generate questions that are not overly complex, we restrict the ontologies to be linear"
>
> Our paper and the rebuttal already provide a detailed explanation of the generation process. Regarding structural characteristics, as addressed in our response to Reviewer NAgL, our dataset encompasses problems with highly diverse reasoning chains.
>
>
> |    | Easy | Medium | Hard |
> |-----------|------|:-------|------|
> | # Unique logic skeletons | 85   | 221    | 494  |
> | Total number             | 500  | 500    | 500  |
>
> As a comparison, other datasets do not have such diverse reasoning chains (data from FOLIO[1]).
>
> |   | RuleTaker | ProofWriter | FOLIO |
> |--------------------------|------|:-------|------|
> | # Unique logic skeletons | 101   | 101    | 76  |
> | Total number             | 500k  | 500k    | 1435  |
>
> Again, BoardGameQA is not an FOL dataset, making direct comparisons less appropriate.
>
> [1] Han, Simeng, et al. "Folio: Natural language reasoning with first-order logic." arXiv preprint arXiv:2209.00840 (2022).
>
> > Follow-up Q3: As mentioned above, work such as ProntoQA and RuleTaker all come with source code for generation, where one can also customize for complexity.
>
> Please see our response for Follow-up Q1.
>
> > Follow-up Q4: Comparing models trained on 1K data versus 5k does not make any sense even if the former is trained for 5 more times. I do not think one can reasonably draw any conclusion from this comparison.
>
> As mentioned in our response to Q5, FOLIO is trained with 1k examples because its training set only contains 1k instance.
> Even without considering FOLIO, the comparison results with proofwriter, which also contains 5k samples, still highlight that our dataset is of high quality.
>
> > Follow-up Q5:
> >
> > Author: Our study shows that LLMs still struggle significantly with complex reasoning tasks, particularly those involving long chains of reasoning and intricate logic.
> >
> > Reviewer oQpi: This is also sufficiently demonstrated in prior work such as ProntoQA, RuleTaker, BoardGameQA, and Proofwriter.
>
> We would like to point out that datasets like ProntoQA, ProofWriter and RuleTaker do not invlove intricate logics (see our response for Follow-up Q1).

---

> ### Author Response · Authors · 2024-12-03
> **Response to Reviewer oQpi (2/3)**
>
> > Follow-up Q6: The effect of distraction in deductive reasoning is also demonstrated in BoardGameQA
>
> BoardGameQA focuses on natural language reasoning with contradictory information, which is a different domain from FOL reasoning. The distractions in BoardGameQA significantly differ from those in FOL reasoning. In BoardGameQA, distractions are contradictory rules, whereas in FOL reasoning, distractions stem from irrelevant or indirectly related rules that are not used in the reasoning chain. Crucially, unlike in BoardGameQA, distractions in FOL reasoning do not contradict the fundamental premises.
> Below is an example to illustrate the differences:
>
>
> |                | Premises                                                                        | Distractions                                                                     |
> |--------|------------|---------------|
> | BoardGameQA    | All travelers entering Y from X need to show negative covid19 test results.     | Travelers visiting Y for less than a month do not require covid19 tests anymore. |
> | ProverGen  | If Sawyer is a performer, then he either has good dance skills or is a musician | If someone either sings opera or plays an instrument, then they are a musician   |
>
> > Follow-up Q7: A verified training dataset could be a good contribution, though I'm not very confident that checking 60 samples would be sufficient to verify the dataset.
>
> In our paper, we demonstrated that finetuning on the training set enhances the performance of LLMs on both in-distribution and out-of-distribution datasets. This improvement is indicative of the high quality of the training dataset. Should the dataset be of poor quality, performance gains on verified and out-of-distribution datasets would not be observed. Additionally, a manual inspection of 60 randomly selected samples revealed no quality issues, suggesting an error probability of less than 1/60. This further substantiates the robustness and reliability of our dataset.
>
> > Concern 1: It is still unclear what is unique about this dataset. While it seems to be created in a slightly different way than prior work, it is unclear what can people learn from ProverGen about the LLM's reasoning capability that benchmarks fail to reveal as I replied above.
>
> As mentioned in our paper, the main contribution is the framework, which is able to create FOL reasoning dataset that has the following four merits: Scalability; Natural and Diverse Language; Symbolic Representations; Faithful Reasoning Chains.
>
> Our framework's novelty has been thoroughly explained in our previous responses. If they are still not convincing, we can further clarify this aspect through our results.
>
> - We created a dataset that is significantly harder than previous datasets. As mentioned in your review: "Nevertheless, an evaluation dataset could also be a concrete contribution, but in order to be significant, it needs to reveal something new that was neglected and overlooked in other benchmarks, or bring new or significant harder challenges to the table. ", this could be a concrete contribution.
> - Our dataset is devoid of the data contamination issues observed in prior manually created datasets and is designed to evolve alongside advancements in LLMs. This dataset can be updated regularly at minimal cost, effectively preventing data contamination. Additionally, it is model-agnostic, facilitating seamless integration with emerging models as they become available.
> - Our dataset is the first to comprehensively address the essential criteria (e.g., scalability; natural and diverse language; symbolic representations; faithful reasoning chains) for an effective benchmark in FOL reasoning. It can support a bunch of downstream tasks, such as NL-FOL translation, and tool-based logic problem solving.
> - Training on the created dataset enhances the performance of LLMs on both in-distribution and out-of-distribution datasets. This improvement is not observed with previous datasets, such as ProofWriter.

---

> > ### Author Response · Authors · 2024-12-03
> > **Response to Reviewer oQpi (3/3)**
> >
> > > Concern2-Part1: Also, my concern that "the LLM translated facts and rules that make the statements semantically more natural and diverse; it is not sufficiently demonstrated how this aspect contributes to the novelty of this benchmark" was not addressed.
> >
> > Yes, solely using LLMs to translate facts and rules can not be treated as novel. However, as stated above, it is the framework that is novel (Please see our response to Follow-up Q1 and Follow-up Q2).
> >
> > LLMs is a natural choice in our framework. As the reasoning chains in ProverGen are much more diverse and complex than existing benchmarks, which makes prior methods that use templates to translate logic problems impossible. That's why we use LLMs here. Also, using LLMs could enable a more diverse and natural dataset.
> >
> > > Concern2-Part2: Concretely, while the background story was generated by LLM, it does not align with the chain rule tree generated other than providing a context for rule and fact translation.
> >
> > The background story serves a crucial role in our framework by providing context, which enriches the LLM's output and prevents repetitive responses when applying the same rule. Please our response to W1 for Reviewer 64s6.
> >
> > > Concern2-Part3: This dataset is effectively still synthetic as the reasoning chains are not drawn from real-world natural language distribution
> >
> > A dataset can be either manually created or synthesized, both have their own merits and drawbacks. Compared to manually crafted datasets, it avoids data contamination problems and can be created at a significantly lower cost. Aparting from previous synthetic datasets, our dataset features more diverse and natural language, intricate logics, symbolic representations, faithful reasoning chains, and controllable complexity, enhancing its value.

---

### Official Review · Reviewer_64s6 · 2024-10-30

**Soundness:** 4
**Presentation:** 4
**Contribution:** 4
**Rating:** 8
**Confidence:** 4

**Summary:**

The paper proposes an automatic framework for generating high-quality datasets that adhere to First-order Logic principles, while also being scalable and diverse. The pipeline consists of three stages: Background Story Generation, Logic Skeleton Generation, and Statement Translation. Additionally, the framework introduces distracting premises into the dataset to enhance the comprehensiveness of the benchmark. Experiments show that state-of-the-art LLMs struggle with these logical reasoning tasks, and fine-tuning LLMs on this dataset leads to greater improvement compared to previous logical reasoning benchmarks. An ablation study is conducted to demonstrate the necessity of including distracting factors.

**Strengths:**

I also work on logical reasoning, and I personally really like this paper. It addresses some key limitations of the previous benchmarks.

(1)	For the first time, this paper proposes an automatic pipeline that fully encompasses First-order Logic (FOL) relationships while being scalable and faithful. Additionally, it offers more natural and diverse language compared to previous benchmarks, as well as symbolic language, and includes a complete reasoning chain.

(2)	Experiments demonstrate that this benchmark poses a significant challenge to state-of-the-art LLMs.

(3)	Fine-tuning using this new dataset results in greater improvements compared to previous logical reasoning datasets, highlighting the advantages of the dataset.

(4)	It reduces the issues of disconnection and contradiction present in previous benchmarks, making the evaluation more reliable.

**Weaknesses:**

I don't see any major weaknesses in this paper, but it could benefit from improvements in the following areas:

(1)	Some details require further clarification. When generating the logic skeleton, how are the facts and rules selected? Are they extracted from the generated background story? Providing more details on how the FOL relationships are incorporated at this step would help the reader better understand the process.

(2)	You mentioned that the bottom-up approach in previous work may result in disconnected and contradictory premises. This is a crucial point, as I have also encountered such cases while working on symbolic reasoning datasets. Is it possible to verify this claim with some data? For example, you could show the proportion of disconnected and contradictory cases in datasets like ProofWriter. I understand that, intuitively, the top-down method should generate fully connected logic. So alternatively, you could provide a qualitative analysis of why prior methodology tends to have this problem and how your method effectively addresses it.  Besides, regarding the issue of self-contradictory premises, since you include distracting premises, how do you ensure that these distractions do not lead to contradictory conclusions? While you mention that the distractions do not directly affect the core premises needed for the final conclusion, could they potentially introduce indirect contradictions?

(3)	It might be helpful to emphasize the importance of reducing disconnected and contradictory cases in your contribution, as they hinder the reliability of the evaluation.

(4)	Given the probabilistic nature of LLMs, the benchmark could be further improved by implementing a quality control process, particularly in the stages of generating the logic skeleton and translating FOL into natural language.

**Questions:**

(1)	How are the facts and rules selected when generating the logic skeleton, and how do you incorporate the full set of FOL relationships at this stage?

(2)	Does the LLM introduce any errors when translating FOL statements into natural language, as well as during the Logic Skeleton Generation?

(3)	Is there any invalid case generated and is there a quality control process in place to filter out invalid cases?

---

> ### Author Response · Authors · 2024-11-21
> **Response to Reviewer 64s6 (1/2)**
>
> Thank you for your constructive feedback and support for our work. We are grateful that you are interested in our paper. We respond to your questions and suggestions below.
>
> > W1: Some details require further clarification. When generating the logic skeleton, how are the facts and rules selected? Are they extracted from the generated background story? Providing more details on how the FOL relationships are incorporated at this step would help the reader better understand the process.
>
> Thank you for your suggestion. We appreciate the opportunity to clarify the process.
>
> When generating the logic skeleton, neither facts nor rules are directly extracted from the background story. Instead, the rules are randomly sampled from combinations of 1-2 connectives (see details in our response to Q5 of Reviewer NAgL). Each sampled rule is validated with Prover9 to ensure its capability to deduce the required conclusion effectively.
>
> In contrast, the facts are generated during the Statement Translation by leveraging LLMs. We prompt the LLMs to replace placeholders in the logic skeleton with suitable predicates, while ensuring not to contradict with real-world common sense at the same time. See the prompt in Appendix B.
>
> The primary function of the background story in our framework is to establish the context for the problem. This context guides the LLMs in promoting diversity in the responses. Without this contextual backdrop, LLMs might fail to produce diversified outputs, a phenomenon observed in other domains as well [1].
>
> We hope this explanation clarifies the process. We have incorporated this clarification into our revised paper.
>
> [1] Jentzsch, Sophie, and Kristian Kersting. "ChatGPT is fun, but it is not funny! Humor is still challenging Large Language Models." arXiv 2023.
>
> > W2 part1: You mentioned that the bottom-up approach in previous work may result in disconnected and contradictory premises. Is it possible to verify this claim with some data?
>
> Thank you for your insightful questions. We provide a qualitative analysis of why the prior methodology tends to encounter issues with disconnected or contradictory premises.
>
> The bottom-up approach needs to prepare all the facts at the beginning and then merge them carefully using some rules. This process can be challenging as it may encounter some facts that are hard to merge. In the table presented below, the process initiates with F1 (practice_piano) and F2 (healthy_lifestyle). After the first iteration, we encounter challenges in merging F3 (improved_skills) and F4 (fewer_health_problem).
>
> | Rules                                              | Natural language                                                                |
> |----------------------------------------------------|---------------------------------------------------------------------------------|
> | F1 (practice_piano) → F3 (improved_skills)         | If Jack practices piano every day, then his skills will greatly improve         |
> | F2 (healthy_lifestyle) → F4 (fewer_health_problem) | Anyone who has a healthy lifestyle is likely to experience fewer health problem |
>
> In contrast, our top-down method starts with a final goal and back-propagate it consecutively, ensuring each step naturally connects to the next. This approach prevents disconnection by maintaining a clear line of reasoning throughout the process. Moreover, by initiating from the goal, this approach facilitates a more controlled distribution of the goal's truth value.
>
> > W2 part2: Besides, regarding the issue of self-contradictory premises, since you include distracting premises, how do you ensure that these distractions do not lead to contradictory conclusions? While you mention that the distractions do not directly affect the core premises needed for the final conclusion, could they potentially introduce indirect contradictions?
>
> In our framework, we employ two types of distractions. The first type of distractions are created by altering subject names. Since these names are generic (e.g., common human or animal names without specific meaning), they do not inherently introduce contradictions. The second type of distractions arise during the logic skeleton generation stage. Our symbolic prover ensures that such distractions only lead to an "Uncertain" status for related facts in the reasoning tree (e.g., f1, f4 in Figure 1), thereby protecting the integrity of the core premises.
>
> While we acknowledge the theoretical possibility of indirect contradictions, our manual review of 60 examples (20 examples from each subset) from our dataset revealed no such instances. This suggests that the likelihood of potential indirect contradictions is low, approximately less than 2% based on our sample.
>
> We are committed to further investigating this issue and will incorporate more robust checks in future iterations of our work. Thank you once again for highlighting this important consideration.

---

> ### Author Response · Authors · 2024-11-21
> **Response to Reviewer 64s6 (2/2)**
>
> > W3: It might be helpful to emphasize the importance of reducing disconnected and contradictory cases in your contribution, as they hinder the reliability of the evaluation.
>
> Thank you for your advice. We have incorporated it into the contributions in our paper.
>
> > W4: Given the probabilistic nature of LLMs, the benchmark could be further improved by implementing a quality control process, particularly in the stages of generating the logic skeleton and translating FOL into natural language.
>
> Thank you for your insightful feedback. We appreciate the opportunity to clarify our quality control processes.
>
> 1. **Logic Validation:** Our framework includes a robust logic validation step. For each instance, we input both core premises and distractions into the symbolic prover to ensure they correctly deduce the conclusion's truth value.
> 2. **Conflicts Resolution:** During rule translation, we check for previously used predicates to avoid redundancy and potential conficting facts. Additionally, we utilize LLMs to assess whether the generated universal rules align with real-world knowledge. In cases of conflict, we opt for specific rules instead.
> 3. **Translation Quality Control:** We apply a heuristic method to ensure that all involved entities appear in both symbolic expression and natural language expression. For example, when translating "poet(Sawyer)" into "Sawyer is a poet," we verify that both the name "Sawyer" and the predicate "poet" are present in the translation.
>
> Additionally, the training part of the paper also serves as quality checking. Finetuning on the generated dataset enhances the performance of LLMs on both in-distribution and OOD datasets, indicating the relatively high quality of the generated data.
>
> Despite the above quality control processes and finetuning experiments, we agree that our framework can be further improved by introducing more advanced quality control processes. We will explore more about it in our future work.
>
> > Q1: How are the facts and rules selected when generating the logic skeleton, and how do you incorporate the full set of FOL relationships at this stage?
>
> Please see our response to W1. For the coverage of FOL rules, please see our response to Q5 of reviewer NAgL.
>
> > Q2: Does the LLM introduce any errors when translating FOL statements into natural language, as well as during the Logic Skeleton Generation?
>
> Regarding the Logic Skeleton Generation, no errors are introduced as this process is entirely handled by the symbolic prover, ensuring its accuracy. However, during the translation of FOL statements into natural language, there is a possibility that the LLM may introduce errors. To mitigate this, as stated in our response to W4, we have implemented heuristic quality control processes to minimize errors.
>
> To further evaluate the quality of our generated data, we performed a manual inspection (same as in our response to W2 part2) and did not find any instance with translation errors. This suggests that the likelihood of potential translation errors, estimated to be less than 2%.
>
> > Q3: Is there any invalid case generated and is there a quality control process in place to filter out invalid cases?
>
> Please see our response to W4 and Q2.
>
> We greatly appreciate the reviewer's feedback and welcome further discussions.

---

> > ### Comment · Reviewer_64s6 · 2024-11-22
> >
> > Hi authors, thank you for your detailed response. I am satisfied with your explanation and will retain my rating.
> >
> > I have a clarification question: Is all the necessary knowledge explicitly listed in the premises to arrive at the desired answer, without requiring the interpretation of commonsense or assumptions? In other words, does the dataset disentangle the target FOL reasoning from commonsense reasoning?

---

> > > ### Author Response · Authors · 2024-11-22
> > > **Response to Reviewer 64s6**
> > >
> > > Thank you for your quick response. Yes, the dataset is designed such that all necessary information is explicitly provided in the premises. This ensures that the target FOL reasoning can be performed without relying on commonsense knowledge or additional assumptions.

---

### Official Review · Reviewer_BNmr · 2024-11-03

**Soundness:** 3
**Presentation:** 3
**Contribution:** 3
**Rating:** 6
**Confidence:** 4

**Summary:**

The authors propose a first-order logic reasoning data generation pipeline and introduce a reasoning benchmark/dataset called ProverGen. This pipeline combines symbolic provers with LLMs to generate scalable, diverse, and high-quality data. Mainstream LLMs are evaluated on this benchmark. Additionally, the authors trained a Llama3.1-8B model on a training set produced by the pipeline, and results show that ProverGen enhances both in-distribution and out-of-distribution logic reasoning abilities.

**Strengths:**

1. The paper is well-written with clear presentation. The authors effectively explain how ProverGen differs from existing FOL reasoning benchmarks.

2. The main contribution is a novel top-down generation framework that creates logically coherent and diverse FOL problems, includes varying difficulty levels and two types of distractions, poses meaningful challenges, as SOTA models achieve only 50% accuracy on hard problems.

3. The experiements are clear and relatively sound. The authors evaluates multiple SOTA LLMs with both standard and chain-of-thought prompting, demonstrates improvement through model finetuning, shows generalization to out-of-distribution FOL tasks.

4. Provides reproducible results and promises to release code and dataset.

**Weaknesses:**

1. The scope is relatively limited. Focuses exclusively on first-order logic reasoning, which may not fully represent real-world reasoning scenarios. Lacks evaluation on general reasoning benchmarks (e.g., MMLU, GSM8K, BIG-bench) to assess broader impact of training.

2. While in-distribution performance shows significant improvement after finetuning (>30% increase), out-of-distribution gains are marginal (5-8%). The modest OOD improvement suggests that training on ProverGen may not substantially enhance general reasoning capabilities. Questions remain about whether the skills learned from this benchmark can transfer to broader reasoning tasks.

3. Lacks detailed analysis of domain distribution in the generated dataset. The diversity of the generated data is not fully revealed.

**Questions:**

1. Please discuss the points mentioned in the weakness section.

2. In the case study section, are there any statistics showing how many cases improved with the help of training?

3. Would changing the generation model from Llama3.1-70B-Instruct to a more advanced model make any difference?

4. Is this generation pipeline only helpful for building benchmarks, or can the synthetic dataset be used for future training?

---

> ### Author Response · Authors · 2024-11-21
> **Response to Reviewer BNmr (1/2)**
>
> Thank you for your comments and constructive suggestions. We address your concerns and questions below.
>
> > W1: The scope is relatively limited. Focuses exclusively on first-order logic reasoning, which may not fully represent real-world reasoning scenarios. Lacks evaluation on general reasoning benchmarks (e.g., MMLU, GSM8K, BIG-bench) to assess broader impact of training.
>
> Thank you for your thoughtful feedback. Aligned with previous works[1-3], we primarily focused on FOL because it is a fundamental aspect of deductive reasoning. Our results show that LLMs still face challenges with complex FOL tasks, particularly in scenarios involving long-chain reasoning. While benchmarks such as MMLU, GSM8K, and BIG-bench are valuable for evaluating general reasoning abilities, they differ significantly from FOL reasoning.
>
> Addressing task generalization across a broader scope, as you suggest, would require systematically addressing challenges like catastrophic forgetting. This would involve preparing a comprehensive set of reasoning-related instruction-tuning data, optimizing data-mixing strategies, and conducting controlled experiments to assess whether incorporating our generated FOL data enhances performance across a broader range of reasoning tasks.  Such investigations fall outside the scope of our current research. However, we appreciate this insightful question and acknowledge the importance of broader benchmarks, we seriously consider it and plan to explore it in future studies.
>
> [1] Olausson, Theo X., et al. "LINC: A Neurosymbolic Approach for Logical Reasoning by Combining Language Models with First-Order Logic Provers." EMNLP 2023.
>
> [2] Yang, Yuan, et al. "Harnessing the power of large language models for natural language to first-order logic translation." arXiv preprint arXiv:2305.15541 (2023).
>
> [3] Kimura, Daiki, et al. "Neuro-Symbolic Reinforcement Learning with First-Order Logic." EMNLP 2021.
>
> > W2: While in-distribution performance shows significant improvement after finetuning (>30% increase), out-of-distribution gains are marginal (5-8%). The modest OOD improvement suggests that training on ProverGen may not substantially enhance general reasoning capabilities. Questions remain about whether the skills learned from this benchmark can transfer to broader reasoning tasks.
>
> It is a normal phenomenon that the performance improvements on in-distribution datasets are larger than OOD datasets. However, ProverGen uniquely enhances OOD performance compared to datasets like ProofWriter and FOLIO. As recognized by Reviewer NAgL and 64s6, finetuning on ProverGen improves OOD reasoning on datasets such as FOLIO and PrOntoQA.
>
> > W3: Lacks detailed analysis of domain distribution in the generated dataset. The diversity of the generated data is not fully revealed.
>
> Thank you for your feedback. We have conducted a detailed analysis of the domain distribution within the ProverGen dataset. For each instance, predicates from premises were extracted and categorized using WordNet. The domain of each instance was determined by the majority category of its predicates. Our analysis reveals that the dataset spans a wide range of domains. This diversity ensures broad applicability and robustness in various contexts.
>
>
> | category      | count |
> |---------------|-------|
> | possession    | 237   |
> | cognition     | 219   |
> | social        | 215   |
> | communication | 201   |
> | stative       | 168   |
> | change        | 100   |
> | creation      | 58    |
> | contact       | 46    |
> | motion        | 45    |
> | perception    | 36    |
> | consumption   | 33    |
> | competition   | 24    |
> | emotion       | 19    |
> | body          | 12    |
> | weather       | 1     |

---

> ### Author Response · Authors · 2024-11-21
> **Response to Reviewer BNmr (2/2)**
>
> > Q1: Please discuss the points mentioned in the weakness section.
>
> Please see our responses above.
>
> > Q2: In the case study section, are there any statistics showing how many cases improved with the help of training?
>
> We have counted the number of improved cases after training in each dataset. See the table below.
>
>
> |               | ProverGen-Easy | ProverGen-Medium | ProverGen-Hard | ProntoQA | ProofWriter | FOLIO |
> |---------------|:---------------|:-----------------|:---------------|:---------|-------------|-------|
> | # Improvement   | 107            | 220              | 173            | 44       | 53          | 7     |
> | Test Set Size | 500            | 500              | 500            | 500      | 600         | 140   |
>
> > Q3: Would changing the generation model from Llama3.1-70B-Instruct to a more advanced model make any difference?
>
> We conjecture that changing the generation model from Llama3.1-70B-Instruct to a more advanced model would make a difference. Our framework is designed to be model-agnostic, allowing seamless integration of newer and more capable models as they become available. This adaptability means that as model capabilities advance, our dataset can benefit from enhanced quality and diversity of generated data.
>
> > Q4: Is this generation pipeline only helpful for building benchmarks, or can the synthetic dataset be used for future training?
>
> The generation pipeline we propose is indeed beneficial for benchmarking as well as training. As demonstrated in Section 5 of our paper, the synthetic datasets we create can be effectively utilized for training, leading to improved performance on both in-distribution and OOD evaluations.
>
> We greatly appreciate the reviewer's feedback and welcome further discussions.

---

> > ### Comment · Reviewer_BNmr · 2024-11-23
> >
> > Thank you to the authors for the reply and the extra analysis and statistics provided. I am willing to offer a raise in the "contribution" rating, but the overall rating will remain the same.

---

### Official Review · Reviewer_NAgL · 2024-11-04

**Soundness:** 3
**Presentation:** 4
**Contribution:** 3
**Rating:** 8
**Confidence:** 4

**Summary:**

The authors present a new method to generate first-order reasoning QA data to evaluate the reasoning abilities of LLMs. For each example, they utilize a synthetic generative process to produce the skeleton of the proof tree. They use an LLM to generate the subject and predicates of the example, where a seed subject and topic are sampled from public datasets of names and WordNet, respectively. The use of the LLM facilitates the generation of logical facts and rules that are consistent with the real-world and ensures the generated natural language sentences are linguistically diverse and realistic, in contrast with other synthetic data generation pipelines which use templates. Their generated dataset, called ProverGen, is demonstrated to be effective in measuring the reasoning abilities of LLMs.

Interestingly, the authors also show that fine-tuning models on ProverGen improves their OOD reasoning capabilities on datasets such as FOLIO and PrOntoQA.

**Strengths:**

- The authors present a pipeline approach for generating QA data to test the reasoning abilities of LLMs.
- The natural language of the text in the examples generated by ProverGen are linguistically diverse and realistic, thanks to the use of an LLM in converting logical form into natural language.
- The authors ensure the proof tree of each example is correct by utilizing a symbolic prover (Prover9).

**Weaknesses:**

- Some of the comparisons to previous work are not completely accurate (see questions below).
- The coverage of the output examples is not well described (i.e., the set of possible output proof skeletons, the set of deduction rules used for generation, etc).
- Since the described rules are specifically designed to be consistent with the LLM's background knowledge, the model can exploit background knowledge as a heuristic when solving examples from ProverGen.

**Questions:**

I provide more detailed questions and comments below. The paper is well-written with only a small number of grammatical errors.

Table 1: While it is true that the provided data CSVs for PrOntoQA doesn't contain logical form annotations for each example, they do provide code to parse any sentence in the dataset into FOL logical forms (since each example was generated by converting logical forms into natural language sentences).

Line 86: "ProverGen’s difficulty setting demonstrates a fine-grained complexity"
  I understand the intended meaning of this phrase thanks to the part of the sentence that follows this phrase, but this phrase itself is somewhat difficult to understand on its own.

It would also be good to mention the disadvantage of data contamination for manually-annotated datasets. In addition, a discussion of how the proposed dataset relates to the problem of data contamination would be welcome.

Footnote 3: PrOntoQA avoids using real-world concepts to ensure the generated sentences do not contradict with real-world knowledge (i.e. the "fictional ontology" setting). They also provide true and false ontologies to specifically LLM behavior on examples that contain facts/rules that are consistent (or inconsistent) with the real-world.

Section 3.2.2: What are the set of possible deduction rules from which proof steps are sampled? Are deduction rules involving universal and existential quantifiers generated? I assume since the authors claim ProverGen has coverage over all connectives and quantifiers in FOL, that there is at least one possible deduction rule for each connective and quantifier. More broadly, what are the completeness properties of this proof skeleton generation procedure? What are the kinds of proofs that can and cannot be generated?

Line 303: "we opt to the specific rule" -> "we opt for the specific rule"

Have the authors experimented with more than 2 few-shot examples in the prompt? If so, were there any significant differences in behavior?

---

> ### Author Response · Authors · 2024-11-21
> **Response to Reviewer NAgL (1/2)**
>
> Thank you for your comments and constructive suggestions. We address specific weaknesses and questions below.
>
> > W1: Some of the comparisons to previous work are not completely accurate (see questions below).
>
> Please see our responses to Q1 and Q4.
>
> > W2: The coverage of the output examples is not well described (i.e., the set of possible output proof skeletons, the set of deduction rules used for generation, etc).
>
> **Proof Skeletons:** We evaluated the number of unique logic skeletons per subset in ProverGen. As reasoning steps increase, the number of unique skeletons also increases, with the hard subset mostly having unique structures.
>
> |                          | easy | medium | hard |
> |--------------------------|:-----|--------|------|
> | # Unique proof skeletons | 85   | 221    | 494  |
> | Total number             | 500  | 500    | 500  |
>
> **Deduction Rules:** Please see our response to Q5.
>
> > W3: Since the described rules are specifically designed to be consistent with the LLM's background knowledge, the model can exploit background knowledge as a heuristic when solving examples from ProverGen.
>
> Our approach involves generating rules by replacing placeholders in logic skeletons with predicates, which are created by LLMs with the guidance of fictional background stories. The names and keywords used for creating background stories are generic, minimizing the risk of "shortcuts".
>
> To further investigate the potential use of shortcuts, we conducted an experiment by removing universal rules from 60 randomly selected instances in ProverGen. We evaluated GPT-4 and Llama-3.1-70B-Instruct on this "corrupted" dataset. If the models were relying heavily on inherent knowledge as shortcuts, their performance would remain roughly unaffected despite the absence of universal rules.
>
> However, our results showed a significant drop in performance, indicating that these models do not heavily rely on background knowledge to solve the problems. This supports our claim that the models are not exploiting background knowledge as heuristics.
>
> |                               | Original | Corrupted | $\Delta$  |
> |-------------------------------|:---------|-----------|--------|
> | GPT-4o-Direct                 | 58.33    | 43.33     | -15.99 |
> | GPT-4o-CoT                    | 68.33    | 45.00     | -23.33 |
> | Llama-3.1-70B-Instruct-Direct | 65.00    | 48.33     | -16.67 |
> | Llama-3.1-70B-Instruct-CoT    | 65.00    | 53.33     | -11.67 |
>
> We have added these results to our revised paper in Appendix C.
> > Q1: While it is true that the provided data CSVs for PrOntoQA doesn't contain logical form annotations for each example, they do provide code to parse any sentence in the dataset into FOL logical forms (since each example was generated by converting logical forms into natural language sentences)
>
> Thank you for pointing it out. We have corrected this and updated Table 1 accordingly in the revised PDF.
>
> > Q2: Line 86: "ProverGen’s difficulty setting demonstrates a fine-grained complexity" I understand the intended meaning of this phrase thanks to the part of the sentence that follows this phrase, but this phrase itself is somewhat difficult to understand on its own.
>
> We apologize for any confusion. We've revised it to:  “ProverGen's difficulty settings are carefully designed to ensure appropriate complexity”.
>
> > Q3: It would also be good to mention the disadvantage of data contamination for manually-annotated datasets. In addition, a discussion of how the proposed dataset relates to the problem of data contamination would be welcome.
>
> Thank you for your insightful suggestion. Data contamination is indeed a significant issue in manually annotated datasets, as it is difficult to update them frequently. This limitation can lead to biased evaluations and hinder true generalization due to potential data leakage.
>
> Our ProverGen framework addresses this challenge by enabling the generation of new datasets using diverse models and controlled complexity. This approach ensures that the datasets remain fresh and uncontaminated, mitigating the problem of data contamination and supporting more reliable and unbiased evaluations. We have included this discussion in the introduction of our revision.

---

> ### Author Response · Authors · 2024-11-21
> **Response to Reviewer NAgL (2/2)**
>
> > Q4: Footnote 3: PrOntoQA avoids using real-world concepts to ensure the generated sentences do not contradict with real-world knowledge (i.e. the "fictional ontology" setting).
>
> Thank you for pointing it out. We have revised Footnote 3 as follows:
>
> "Both ProofWriter and PrOntoQA generate natural language expressions by using predefined templates for each logical rule... PrOntoQA intentionally avoids real-world concepts to prevent conflicts with real-world knowledge. They also evaluate LLMs' behavior on examples that contain facts and rules that are consistent (or inconsistent) with the real-world knowledge. In contrast, ProofWriter does not incorporate such a mechanism."
>
> > Q5: What are the set of possible deduction rules from which proof steps are sampled? Are deduction rules involving universal and existential quantifiers generated? More broadly, what are the completeness properties of this proof skeleton generation procedure?
>
> **Coverage of generated deduction rules:** Our rules encompass seven fundamental FOL symbols, including four connectives (∧, ∨, →, ⊕), two quantifiers (∀, ∃), and negation (¬).  During logic skeleton generation, deduction rules are sampled from combinations of these connectives, as listed below:
> [A → B, A ⊕ B, A ∨ B, A ∧ B,  A ∧ (B ∧ C), (A ∧ B) → C, A ∧ (B ∨ C), A ∨ (B ∧ C), (A ∨ B) → C, A ∨ (B ∨C), (A ⊕ B) → C , A → (B ⊕ C), A→ (B ∨ C), A → (B ∧ C)]
>
> Each rule initially includes ∀. If contradicted by real-life common sense judged by LLM, ∀ is replaced with ∃ to reduce contradictions. Negation (¬) is typically used within the rule’s fact.
>
> **Deduction rules not generated:** (1). Rules with more than two connectives are excluded due to complexity and ambiguity. (2) Rules like [(A ∧ B) ⊕ C, A ∨ (B ⊕C), (A ⊕ B) ∨ C, (A ⊕ B) ∧ C,  A → (B → C)] are also excluded due to ambiguous translations in our practice.
>
> > Q6: Line 303: "we opt to the specific rule" -> "we opt for the specific rule”
>
> We have corrected it in the PDF.
>
> > Q7: Have the authors experimented with more than 2 few-shot examples in the prompt? If so, were there any significant differences in behavior?
>
> We have conducted experiments using 5-shot examples in both Direct and CoT settings. The results are detailed below, with the change compared to 2-shot are presented in parentheses.
>
> **Direct Prompting:**
> |    | ProverGen-Easy | ProverGen-Medium | ProverGen-hard | ProntoQA  | ProofWriter   | FOLIO    | Avg $\Delta$ |
> |----|:----|:----|:-----|:------|-----|-----|:----|
> | GPT-4o      | 85.20 (-2.00)  | 68.40 (-0.20)    | 43.80 (-2.40)  | 88.20 (-3.60) | 57.33 (+1.00) | 72.86 (+5.00) | -0.37   |
> | Claude-3.5-Sonnet   | 93.20 (+8.20)  | 76.80 (+8.60)   | 51.00 (+8.20)  | 85.20 (-3.40) | 50.33 (-4.67) | 74.29 (-3.56) | +2.23   |
> | Llama3.1-8B-Instruct  | 64.40 (+17.80) | 41.80 (-1.20)   | 38.00 (-1.00)  | 58.00 (+7.60) | 44.50 (+0.70) | 46.43 (-7.86) | +2.67   |
> | Llama3.1-70B-Instruct  | 82.80 (+0.80)  | 64.40 (+0.20)   | 49.20 (+1.60)  | 74.00 (-6.60) | 51.33 (+1.00) | 69.29 (+1.43) | -0.26   |
> | Mistral-7B-Instruct  | 57.40 (+0.60)  | 48.40 (+1.60)   | 37.40 (+0.20)  | 52.40 (+2.40) | 47.83 (+5.50) | 52.86 (-1.43) | +1.48   |
> | Mistral-Large-Instruct | 85.00 (+0.40)  | 71.40 (+2.20)    | 53.6 (+4.00)   | 67.40 (-3.60) | 63.50 (+3.17) | 76.43 (-0.71) | +0.91  |
> | Mixtral-8x22B-Instruct | 77.60 (+2.20)  | 58.00 (+0.60)   | 39.60 (+0.60)  | 60.00 (-5.20) | 40.50 (+0.33) | 73.57 (-0.72) | -0.37  |
>
> **CoT Prompting:**
>
> |  | ProverGen-Easy | ProverGen-Medium | ProverGen-hard | ProntoQA  | ProofWriter  | FOLIO  | Avg $\Delta$ |
> |-----|:------|:-----|:-----|:-----|:----|----|----|
> | GPT-4o | 95.00 (+0.80) | 81.20 (+1.80) | 55.00 (+5.00)  | 99.20 (-0.80) | 71.00 (+3.67) | 74.29 (+2.15) | +2.10    |
> | Claude-3.5-Sonnet  | 92.00 (-3.20)  | 81.60 (-2.00)    | 61.00 (+4.60)  | 95.80 (-3.40) | 75.50 (-0.83) | 82.86 (+2.15) | -0.45    |
> | Llama3.1-8B-Instruct   | 80.60 (+5.00)  | 45.40 (-1.20)  | 31.20 (-2.40)  | 74.80 (-4.80) | 58.67 (+1.84) | 57.86 (-5.68) | -1.21    |
> | Llama3.1-70B-Instruct  | 92.60 (+2.20)  | 73.40 (+0.20)  | 50.20 (+3.40)  | 92.80 (-2.60) | 67.00 (-4.17) | 75.00 (+0.71) | -0.04    |
> | Mistral-7B-Instruct    | 65.60 (-6.40)  | 50.20 (-0.80) | 38.00 (-3.80)  | 65.80 (+4.60) | 48.33 (+2.33) | 63.57 (-0.01) | -0.68    |
> | Mistral-Large-Instruct | 94.40 (+1.80)  | 77.60 (+1.80) | 56.60 (+4.40)  | 99.00 (+0.40) | 77.67 (+4.17) | 82.86 (-0.71) | +1.98    |
> | Mixtral-8x22B-Instruct | 91.00 (+3.40)  | 73.80 (+7.00)  | 50.20 (+2.60)  | 86.20 (+6.60) | 59.67 (+2.00) | 72.14 (-1.43) | +3.36    |
>
> Our findings indicate that increasing the number of examples from 2-shot to 5-shot did not consistently enhance the performance of LLMs. This variability in outcomes may be attributed to differences in the models' in-context learning capabilities, which can affect how effectively they utilize additional examples. We have added the results in Appendix G.
>
> We greatly appreciate the reviewer's feedback and welcome further discussions.

---

> ### Comment · Reviewer_NAgL · 2024-11-25
> **Response to Submission2601 Authors**
>
> I appreciate the authors' timely and thoughtful response.
>
> 1. On experiments with corrupted dataset without universal rules:
>
> That makes sense that if the generated proofs are sufficiently generic, the model would be less likely to exploit background knowledge to use shortcuts when reasoning. Though in principle it is possible (but I think it's unlikely) that "real world" proofs about specific topics would be correlated with specific proof structures or patterns, which the model may be able to learn and exploit.
>
> The experimental results on the "corrupted" examples where universal rules are removed look interesting. Though I wonder if there is a confounding effect, for example, by not having any examples of universally-quantified rules in the prompt, the model may be less inclined to use modus ponens at all. One way to control for this would be to add examples of universally-quantified rules that are irrelevant. However, even this experiment would have potential confounders where the presence of irrelevant information may negatively affect the model's reasoning ability. In general, it seems rather difficult to test for this effect cleanly. More ideally, there would be a source of rules that are OOD from the model's training (perhaps from a domain or game that was released after the model's training cutoff, or a small human-annotated set of rules, or fictional rules).
>
> 2. On revisions:
>
> I appreciate the authors' revisions that more accurately compare with previous work, and the new discussion of data contamination in the paper.
>
> 3. On the coverage of proof generation:
>
> I thank the authors for their clarification on the coverage of the proof generation. When generating logical forms, are compositional forms generated? (e.g., nested conjunction within a disjunction within a universally-quantified rule) And while I do have a better understanding now about the generative process of logical forms, I am still a bit unclear on what deduction rules (i.e., rules of inference) are generated. Does the dataset focus on modus ponens? Or are other deduction rules also generated? (e.g., such as conjunction rules such as given A is true and B is true, conclude that A & B is true, or rules involving negation, etc)
>
> If only modus ponens is used, then this inference system is better compared to a [Hilbert system](https://en.wikipedia.org/wiki/Hilbert_system) (as opposed to, say, natural deduction), which would be perfectly valid.
>
> 4. On few-shot prompting experiments:
>
> The 5-shot few-shot prompting results look very interesting, and it doesn't seem like there is a significant different difference in performance relative to 2-shot prompting.
>
> In light of the proposed revisions, I will raise my score accordingly.

---

> > ### Author Response · Authors · 2024-11-27
> > **Response to Reviewer NAgL**
> >
> > Thank you for your positive feedback. We are more than willing to provide answers to your questions.
> > > Follow-up Q1: On experiments with corrupted dataset without universal rules
> >
> > Thank you for your insightful analysis. Testing the shortcut problem in a straightforward manner is indeed challenging. Using rules that are out-of-distribution from the model's training data is ideal. However, since LLMs are trained on extensive, non-public datasets that encompass a wide range of real-world knowledge, it's difficult to determine if a rule is truly OOD. Creating fictional rules is a viable alternative, but it's crucial to ensure they don't conflict with the existing knowledge in LLMs. Representing rules and facts with fictional symbols might be a solution, though this shifts the problem from natural language reasoning to symbolic reasoning, which is a different domain.
> > > Follow-up Q2: On the coverage of proof generation. When generating logical forms, are compositional forms generated?
> >
> > Yes, our framework does generate compositional forms, including examples like $\( A \land (B \land C) \)$, $\( A \land (B \lor C) \)$, $\( A \lor (B \land C) \)$, and $\( A \lor (B \lor C) \)$. However, to ensure clarity and maintain interpretability when translating these forms into natural language, we have intentionally limited the coverage to combinations involving up to two connectives. This decision helps mitigate potential ambiguities that can arise with more complex nested structures.
> > > Follow-up Q3: On the coverage of proof generation. Does the dataset focus on modus ponens? Or are other deduction rules also generated?
> >
> > Thank you for your insightful questions. In our framework, we generate a variety of deduction rules beyond modus ponens. Specifically, we include both $\(A \oplus B\)$ and $\(A \lor B\)$. It's important to highlight that $\(A \lor B\)$ can be transformed into $\(\lnot A \rightarrow B\)$, effectively aligning it with modus ponens in terms of inference. The unique rule in our approach is $\(A \oplus B\)$, which is equivalent to $\((A \rightarrow \lnot B) \land (\lnot A \rightarrow B)\)$.
> >
> > We intentionally exclude rules like $\(A \land B\)$ because they simply present two facts rather than facilitate reasoning. However, we incorporate such rules when they serve as goals in FOL problems. For example, if the context of a problem states that both $A$ and $B$ are true, the objective might be to deduce the truth value of $\(A \land B\)$.
> >
> > We truly appreciate your insightful feedback and welcome further discussions.

---

### Author Response · Authors · 2024-11-21
**Updated Manuscript and Response to all reviewers:**

We thank all the reviewers for their valuable feedback and thoughtful comments on our paper. We have uploaded a revised PDF of the paper. All revisions have been clearly marked using red font and formatted as [Revision: xxx] to facilitate easy identification.

---

### Author Response · Authors · 2024-11-30
**Rebuttal summary**

We sincerely appreciate all the reviewers for their thorough evaluations. The constructive suggestions have greatly enhanced the quality of our work, and their positive feedback is especially inspiring for us.

We are pleased to note that reviewers have recognized:

- Our paper's success in addressing key limitations of previous benchmarks, such as unnatural and monotonous language, the absence of symbolic language and complete reasoning chains, and disconnected or contradictory premises (Reviewers NAgL, BNmr, 64s6).
- The novelty of our framework (Reviewers NAgL, BNmr, 64s6), and the clarity and quality of the paper's presentation (all Reviewers).
- The experiments as interesting, clear, and sound (Reviewers NAgL, BNmr).
- The high quality of the ProverGen dataset, along with scalability, diversity, and faithfulness (Reviewers NAgL, BNmr, 64s6).

We have carefully addressed the concerns and questions from each reviewer, providing detailed responses. Based on this valuable feedback, we have revised the paper with the following updates (highlighted in red for easy identification):

- Discussed data contamination issues in manually-annotated datasets and how our framework addresses them, thanks to Reviewer NAgL's insightful suggestion.
- Emphasized the importance of reducing disconnected and contradictory cases, as suggested by Reviewer 64s6.
- Introduced a new section detailing the quality control process in our framework, in response to questions from Reviewer 64s6 and oQpi.

Once again, we would like to thank all the reviewers for their time and efforts in helping us enhance the paper!

---

### Meta-Review · Area_Chair_u1AD · 2024-12-24

**Metareview:**

This paper proposes a method to generate first order reasoning QA dataset to evaluate the reasoning abilities of LLMs. Whether LLMs indeed learn to reason has been an important topic for at least 2-3 years. Here, there are a few main considerations. First, how to truly verify whether a LLM reasons. Second, is the result generally applicable to some other datasets. Third, can the dataset be beneficial to improve LLM reasoning. This paper in general is well written and easy to follow. And technically, this paper does a solid answer to all of the three main considerations above. The proposed method is novel and generative in its nature, which potentially can alleviate the problem of data pollution. Furthermore, the generated proof tree is guaranteed to be correct. And the OOD-results is somewhat encouraging. In terms of the limitations, I would encourage the authors to further expand the dataset and include more comparison with more LLMs and do a more comprehensive analysis with other datasets that also highlight reasoning. The comparison could provide more insights.

**Additional Comments On Reviewer Discussion:**

The most important issue in the discussion period is whether the proposed benchmark is novel. Due to its size, it clearly could have its limitation, and a reviewer (which I agree with) raises this concern. Yet, its proposed method is principled. And its analysis accompanied with its OOD experiments help make a case that this dataset could be generally useful. Because of this, I believe the pros (rather than cons) are more shared among more reviewers.

---

### Decision · Program_Chairs · 2025-01-22

Accept (Poster)